# Harnessing Diversity for Important Data Selection in Pretraining Large Language Models

Chi Zhang[*‡1,3], Huaping Zhong[*2], Kuan Zhang[1], Chengliang Chai[†1], Rui Wang[3], Xinlin Zhuang[3], Tianyi Bai[3], Jiantao Qiu[3], Lei Cao[4], Ju Fan[5], Ye Yuan[1], Guoren Wang[1], and Conghui He[†3]

[1]Beijing Institute of Technology
[2]SenseTime Research
[3]Shanghai Artificial Intelligence Laboratory
[4]University of Arizona
[5]Renmin University of China
{zc315,ccl}@bit.edu.cn , zhonghuaping@sensetime.com , caolei@arizona.edu
, fanj@ruc.edu.cn , heconghui@pjlab.org.cn

## Abstract

Data selection is of great significance in pretraining large language models, given the variation in quality within the large-scale available training corpora. To achieve this, researchers are currently investigating the use of data influence to measure the importance of data instances, $i.e.$, a high influence score indicates that incorporating this instance to the training set is likely to enhance the model performance. Consequently, they select the top-$k$ instances with the highest scores. However, this approach has several limitations. (1) Calculating the accurate influence of all available data is time-consuming. (2) The selected data instances are not diverse enough, which may hinder the pretrained model's ability to generalize effectively to various downstream tasks. In this paper, we introduce Quad, a data selection approach that considers both quality and diversity by using data influence to achieve state-of-the-art pretraining results. To compute the influence ($i.e.$, the quality) more accurately and efficiently, we incorporate the attention layers to capture more semantic details, which can be accelerated through the Kronecker product. For the diversity, Quad clusters the dataset into similar data instances within each cluster and diverse instances across different clusters. For each cluster, if we opt to select data from it, we take some samples to evaluate the influence to prevent processing all instances. Overall, we favor clusters with highly influential instances (ensuring high quality) or clusters that have been selected less frequently (ensuring diversity), thereby well balancing between quality and diversity. Experiments on Slimpajama and FineWeb over 7B large language models demonstrate that Quad significantly outperforms other data selection methods with a low FLOPs consumption. Further analysis also validates the effectiveness of our influence calculation. Our code and data are available at (https://anonymous.4open.science/r/Quad/).

## 1 Introduction

Recently, large language models (LLMs) have significantly advanced the field of artificial intelligence (Zhao et al., 2023; Hadi et al., 2023; Minaee et al., 2024). Due to the unprecedented number of parameters (model size) and the pre-training on huge amount of training data, LLMs are generalizable a broad spectrum of downstream tasks. However, in practice, the computation resources limit both the model size and the volume of data used in pre-training. In this situation, judiciously

---

[*]Equal Contribution.
[†]Correspondence to: {ccl@bit.edu.cn, heconghui@pjlab.org.cn}
[‡]This work was done during the internship at Shanghai Artificial Intelligence Laboratory.

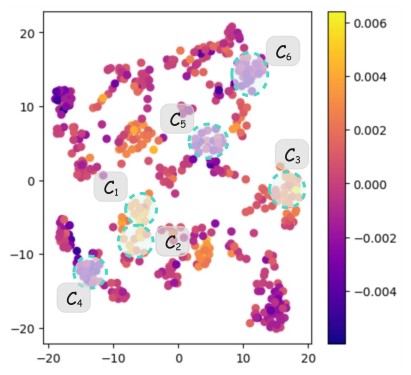

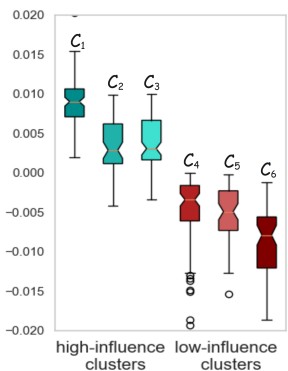

(a) Influence scores of data instances

(b) Influence scores in different clusters

Figure 1: Distribution of influence scores of some sampled data instances.

selecting train datasets is critical for producing highly performance LLMs (Brown, 2020; Du et al., 2022; Gururangan et al., 2020; Zhang et al., 2024a; Hoffmann et al., 2022; Raffel et al., 2020). In particular, the quality of the training datasets vary dramatically, while the LLaMA-3.1 report (Dubey et al., 2024) shows that the use of high quality data in later training stages can greatly improve model performance.

Typical straightforward data selection approaches include rule-based data filtering to clean the data (Raffel et al., 2020; Rae et al., 2021), querying high-performance models (*e.g.*, GPT-4) (Wettig et al., 2024; Sachdeva et al., 2024), surrogate models (Lin et al., 2024; Shao et al., 2024), etc. Although these methods have achieved success on some datasets and models, they rely on simple heuristics to select training data. Without explicitly measuring the impact of the selected data on the model, these methods tend to produce sub-optimal pretraining results. To address this issue, some researchers (Xia et al., 2024; Chai et al., 2023; Yu et al., 2024) start evaluating each data instance by assigning it a score that reflects its impact on the model. Frequently used scoring methods include the influence function (Xia et al., 2024; Liu et al., 2021), early loss (Albalak et al., 2023; Deng et al.), and perplexity (Chen et al., 2024). Among these methods, the influence function consistently delivers state-of-the-art results by effectively approximating the impact of adding each instance to the training set. A higher score signifies a higher priority for selecting a data instance, and hence the top-$k$ (or gumble top-$k$) instances with the highest scores are chosen (Xie et al., 2023; Wettig et al., 2024; Yu et al., 2024).

However, the above methodologies have the following limitations.

**Prohibitive Computation Cost.** First, accurately calculating the influence score of one data instance is expensive, because it involves the computation of the Hessian matrix. However, in the LLM pre-training, the number of the candidate data instances is extremely large. It is thus prohibitively expensive to compute the scores for all of the candidates.

**Lack of Diversity.** Second, assume that all influence scores have been calculated, as shown in Figure 1a. We can see that the top-$k$ instances (*e.g.*, some high-score instances in $C_1$) tend to be closely distributed in the feature space because the influence computation is closely related to the data features. That is, the training instances selected in this way are lack of diversity (*e.g.*, other instances in $C_3$ with high influence are also worth selecting), while as confirmed by some studies (Abbas et al., 2023; Tirumala et al., 2023), diversifying training samples mitigates overfitting, thereby enhancing the generalizability of the model. Therefore, an effective data training selection method should take both the influence scores and the diversity into consideration.

We thus propose `Quad`, a scalabe and effective data selection approach, which successfully addressing above challenges, achieves state-of-the-art pretraining results. Initially, `Quad` organizes the given dataset into clusters where the data instances within each cluster are similar, and those in different clusters exhibit diversity. Hence, we can sample a data subset from a cluster to estimate the accurate average influence of the cluster, so as to represent the cluster quality (e.g. domain relevance) $w.r.t$ the model performance.

Next, leveraging the property of the attention-based Transformer architecture which is widely adopted by the LLMs, we design a novel method to accurately compute the influence of an instance

on LLM pre-training. More specifically, rather than solely relying on the MLP layers to compute the influence (Koh & Liang, 2017; Yu et al., 2024; Grosse et al., 2023; Engstrom et al., 2024), we incorporate the attention layers such that the influence computation considers more semantic information. In addition, given that calculating the Hessian matrix is time-consuming, particularly for attention layers with complex interactions, we incorporate the Kronecker product to approximate the Hessian matrix, thereby greatly expediting the computation. This successfully addresses the computation cost challenge.

To improve diversity, we apply the Multi-Arm Bandit (MAB) technique, where each cluster is regarded as an arm of the MAB. Upon selecting an arm, we draw samples from the cluster to calculate influence scores. Subsequently, `Quad` iteratively samples from clusters, taking into account both the influence score and data diversity, e.g., whether the cluster has already been sampled. Moreover, because this sampling strategy effectively avoids calculating the influence of all instances, it further speeds up the data selection process.

We summarize our main contributions as follows:

- To balance the quality and diversity, we incorporate an iterative MAB solution to first cluster the data instances and select data instances from these clusters.

- We propose a novel method to compute the influence function in attention-based Transformer architecture, so as to precisely measure the data quality in LLM pre-training.

- Experiments on the widely-used dataset Slimpajama, FineWeb and 16 popular downstream tasks demonstrate that `Quad` significantly outperforms state-of-art data selection methods by 1.39% in zero-shot accuracy (train and test on 1.3B and 7B models respectively), also with low computation resources consumption.

## 2 RELATED WORK

**Rule-based Methods.** Initially, researchers often relied on intuition to design hand-crafted heuristics (Soldaini et al., 2024) and (Penedo et al., 2023), aiming to improve data quality. Deduplication is another typical approach for selecting pretraining data, such as (Penedo et al., 2023) and SemDedup (Abbas et al., 2023) which use keyword-based and semantic deduplication, respectively. Additionally, certain approaches employ $n$-gram similarity (Gao et al., 2020; Xie et al., 2023) to assist in choosing corpora that is semantically aligned with the validation set data (Chai et al., 2022). Although these methods effectively filter out noise and redundant data from web sources, they rely on simple heuristics and cannot be well generalized.

**LLM As a Selector.** Although large models such as GPT-4 can effectively assess data quality due to their semantic comprehension capacity, the metrics utilized to rate data ($e.g.$, writing style, educational value etc.) heavily rely on human intuition (Wettig et al., 2024; Penedo et al., 2024; Zhang et al., 2024b; Gunasekar et al., 2023; Peng et al., 2025). This often leads to a mismatch between the selected data and the data desired by the model.

**Surrogate Models.** DeepSeekMath (Shao et al., 2024) proposes an active learning strategy to train a web data classifier. Similarly, in MATES (Yu et al., 2024), a surrogate model was developed to estimate the influence scores of the data instances. RHO-1 (Lin et al., 2024) used a surrogate model trained with high-quality data to perform token-level data filtering. However, these surrogate models are not trained over large-scale data, and thus their generalization ability is limited.

**Perplexity** serves as a metric for selecting high-probability data in a language model. In (Chen et al., 2024; Marion et al., 2023; Muennighoff et al., 2024; Wenzek et al., 2019), perplexity (PPL) is utilized to filter data. As also discussed in Qurating (Wettig et al., 2024), we observe that this method often incorporates a significant amount of simple and redundant data, because they are easy for the model to predict.

**Influence Function** (Grosse et al., 2023; Choe et al., 2024) demonstrates that influence function can reveal the impact of training data on the performance of large models. Consequently, LESS (Xia et al., 2024) and MATES (Yu et al., 2024) utilize influence functions for selecting data during the SFT and pretraining phases, respectively. For large models, computing influence functions is computationally expensive. (Grosse et al., 2023). Hence, given the large amount of data handled during pretraining, directly using LESS (Xia et al., 2024) for data selection at this stage poses considerable

difficulties. To overcome this, MATES (Yu et al., 2024) employs a proxy model to approximate the influence score across the full dataset. However, the limited capacity of this small proxy model hinders its ability to provide accurate influence scores. Furthermore, relying on the influence to select data solely often leads to a lack of diversity in the chosen data.

# 3 METHODS

First, we present our problem statement in §3.1. Next, in §3.2, we explain how our method achieves the balance between quality and diversity in selecting pretraining data. Finally, in §3.3, we introduce how we compute the influence with attention layers more accurately and efficiently.

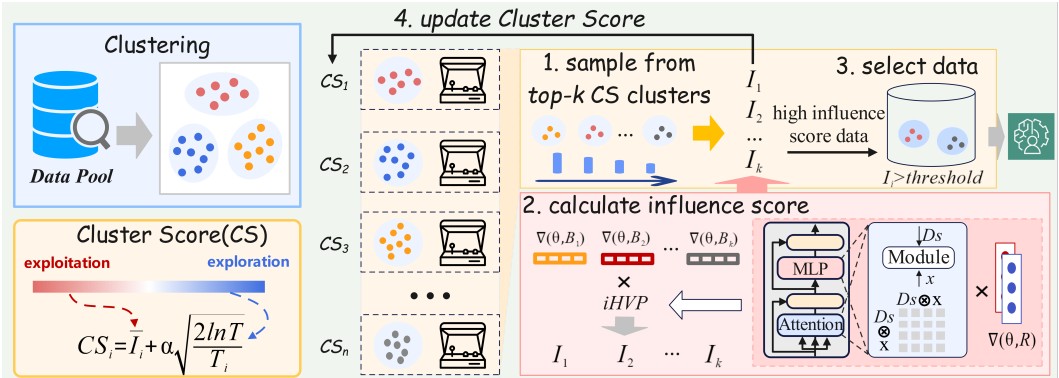

Figure 2: Overview of `Quad`

## 3.1 PROBLEM DEFINITION

In this paper, we study the problem of data selection from a large pool $D_c$ of candidate data for pre-training a large language model (LLM). Formally, given a data pool $D_c$ and a reference dataset $D_r$, the problem is to select a subset $D_b \subset D_c$ to pre-train an LLM $M$, aiming to minimize the loss of the updated model $M'$ on the reference dataset $D_r$. Note that our method is applicable to both pre-training $M$ from scratch, and continuous pre-training scenarios where $M$ starts as a pre-trained checkpoint.

## 3.2 BALANCE BETWEEN QUALITY AND DIVERSITY

As shown in Figure 1b, there are significant variations in the distribution of influence scores among different clusters. To achieve the quality-diversity balance, it is necessary to know the precise average influence score for instances in each cluster. However, Figure 1b shows that the influence scores for each cluster also fluctuate around the average, indicating a certain level of uncertainty. Estimating the average with a small sample size will not be accurate enough, while taking a large number of samples to compute the average influence is costly.

Hence, we propose to use the MAB (Vermorel & Mohri, 2005) technique that is capable of making decisions iteratively under uncertainty. At a high level, each cluster represents an arm of the MAB, and during each iteration, a cluster with a high average influence score tends to be selected and sampled. We will then compute the influence of data instances to update the average. Moreover, clusters that are not visited often present significant opportunities for sampling to balance the diversity.

The overall process of this approach is illustrated in Figure 2. Specifically, our method can be divided into the following four steps: First, we *sample the top-k clusters* with the highest cluster scores (denoted by CS) computed by MAB. Here, the cluster score is determined by both the influence score and the sample frequency. Then we *calculate the influence scores* for the samples in each cluster (Section 3.3). At this point, we *select high scoring samples* to be added for training and use their scores to *update the cluster score* for each cluster. Throughout the iterative process, the MAB algorithm focuses on frequently sampling high-quality clusters that have high influence scores, which also enhances the accuracy of their quality estimation (*i.e.*, updating the average influence $\bar{I}_i$). Simultaneously, it ensures diversity by also sampling less-visited clusters. Next, we discuss how to compute and update the cluster score in details.

**Cluster Score (CS).** The Upper Confidence Bound can effectively balance exploration (*i.e.*, data diversity) and exploitation (*i.e.*, data quality), so we use it as the cluster score to evaluate each cluster, as shown in Equation (1). Specifically, the cluster score is determined by the average influence score $\bar{I}_i$ and the exploration score $\sqrt{\frac{2\ln\sum_j T(C_j)}{T(C_i)}}$, where $T(C_i)$ denotes the frequency of instances sampled from cluster $C_i$, and $\sum_j T(C_j)$ denotes the total times of samples taken from all clusters.

$$CS_i = \bar{I}_i + \alpha\sqrt{\frac{2\ln\sum_j T(C_j)}{T(C_i)}} \tag{1}$$

**Update the cluster score.** During each iteration, a subset of data $B_i$ is sampled from each cluster with a high cluster score (CS). The sum of their influence score $I_i$ can be used to denote the impact of the samples from the cluster $C_i$ on the model.

$$R(C_i)+ = \sum_{z\in B_i} I_\theta(D_r, z), \quad T(C_i)+ = 1 \tag{2}$$

where $R(C_j)$ denotes the total reward accumulated by cluster $C_i$ over several iterations. Then the average influence score $\bar{I}_i$ for cluster $C_i$ can be represented as $\bar{I}_i = \frac{R(C_i)}{T(C_i)}$. As the sample size grows, $\bar{I}_i$ for each cluster $C_i$ steadily approaches the exact average influence of the cluster, which can be used to update the cluster scores for all clusters.

**Data selection.** During each iteration, we pick a small proportion($\gamma$) of data instances from selected clusters. We also require that these instances have influence scores higher than the threshold $\tau$, otherwise we will not select them, which are then added into the training dataset.

---

**Algorithm 1:** `Quad` Algorithm

**Input:** Candidate data pool $D_c$, reference set $D_r$, the model $\theta$
**Output:** Selected data $D_b$

1   $\mathcal{C}$ = Cluster($D_c$);
2   **while do**
3     $C_{top\_k}$ = top-$k$ clusters with the highest Cluster Score(CS) ;
4     $B_{top\_k}$ = mini-batchs sampled from $C_{top\_k}$
5     **for** $C_i$ *in* $C_{top\_k}$ **do**
6       $R(C_j)$ += $\sum_{z\in B_i} I_\theta(D_r, z), \quad T(C_j)$ += 1 ;
7     **end**
8     **for** $C_i$ *in* $\mathcal{C}$ **do**
9       $\bar{I}_i = \frac{R(C_i)}{T(C_i)}$ ;
10      **if** $\bar{I}_i > threshold$ **then** $D_b+ = \gamma C_i$;
11    **end**
12    $CS_i = \bar{I}_i + \alpha\sqrt{\frac{2\ln\sum_j T(C_j)}{T(C_i)}}$ ;
13   **end**
14   **return** $D_b$;

---

### 3.3 INFLUENCE CALCULATION WITH ATTENTION LAYERS

Instead of retraining the large model with each data sample $z$, the impact of $z$ on the model $M$ can be estimated by calculating the influence function for each instance. In this section, we extend the influence calculation to multi-head attention layers and provide acceleration techniques.

$$I_\theta(D_r, z) = -\nabla L(\theta, D_r)(H + \lambda I)^{-1}\nabla L(\theta, z) \tag{3}$$

In the above equation, $I_\theta(D_r, z)$ denote the influence function of data $z$ on model $\theta$. $\nabla L(\theta, D_r)$ and $\nabla L(\theta, z)$ denote the gradient of reference set $D_r$ and data $z$, respectively. Since the training of the large model does not often fully converge, resulting in a non-invertible Hessian matrix $H$, a regularization term $\lambda I$ is introduced (Bae et al., 2022). Equation (3) is typically divided into the following two stages to speed up the computation:

1. Approximate the multiplication of the gradient of the validation set $\nabla L(\theta, D_r)$ and the inverse Hessian matrix $H^{-1}$ using the inverse Hessian vector product (iHVP).

2. Compute the dot product between the iHVP and the gradient of each training data point $\nabla L(\theta, z)$.

While this framework can accelerate the computation of the influence function, scaling it up to large language models (LLMs) with massive parameters is still expensive. Hence, K-FAC (Martens & Grosse, 2015; Ueno et al., 2020) can be used to accelerate the iHVP computation by using the Kronecker product to decompose the Hessian matrix.

The K-FAC approximate the parameters of different MLP layer $\theta_1$, $\theta_2$ and $\theta_3$ as independent. That's because, during the gradient computation and update process, there are usually only minimal direct dependencies between the gradients of different MLP layers. This is particularly evident during back propagation, where the weight updates for each MLP layer are primarily influenced by the parameters of that specific layer. Therefore, the influence function $I_{\theta_1,\theta_2,\theta_3}(D_r, z)$ in K-FAC method can be expressed as:

$$I_{\theta_1,\theta_2,\theta_3}(D_r, z) = I_{\theta_1}(D_r, z) + I_{\theta_2}(D_r, z) + I_{\theta_3}(D_r, z) \qquad (4)$$

In attention mechanisms, there exist complex connections between the Query, Key, and Value layers. As the right-upper corner of Figure 3 shows, separately calculate the hessian matrix of Query, Key and Value layers, will miss massive information. Consequently, it is essential to consider the QKV layers as a unified layer $\theta_{qkv}$ when computing the influence function. Therefore, the influence function $I_{\theta_{att}}(D_r, z)$ can be expressed as:

$$I_{\theta_{att}}(D_r, z) = I_{\theta_{qkv}}(D_r, z) + I_{\theta_o}(D_r, z) \qquad (5)$$

Then, as the right-lower corner of Figure 3 shows, by decomposing the Hessian matrix into a

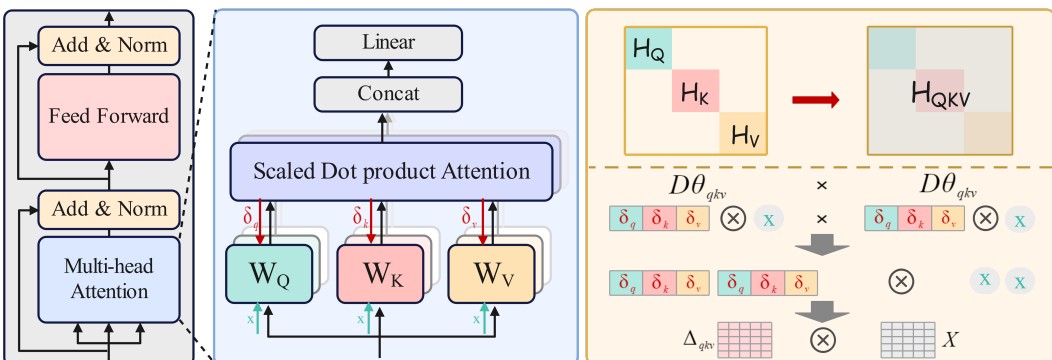

Figure 3: Kronecker Product in calculating iHVP

kronecker product of smaller matrices and computing the inverse of each smaller matrix, we can avoid directly inverting the entire Hessian matrix, significantly reducing computational cost, and accelerate this process:

**Forward Propagation**:

$$Attention(Q, K, V) = softmax(\frac{QK^T}{\sqrt{d_k}})V \qquad (6)$$

**Backward Propagation**:

$$D\theta = vec(DW) = \delta \otimes x \qquad (7)$$

Here, $\otimes$ denotes the Kronecker product, and $vec()$ represents the vectorization operation. Thus, the gradient of $\theta_{qkv}$ can be written as:

$$D\theta_{qkv} = \begin{bmatrix} \mathbf{vec(DW_Q)} \\ \mathbf{vec(DW_K)} \\ \mathbf{vec(DW_V)} \end{bmatrix} = \begin{bmatrix} \delta_q \\ \delta_k \\ \delta_v \end{bmatrix} \otimes x \qquad (8)$$

Let $\delta_{qkv} = \begin{bmatrix} \delta_q \\ \delta_k \\ \delta_v \end{bmatrix}$. Then, the Hessian matrix $H_{qkv}$ can be estimates by:

$$\begin{aligned} H_{qkv} &= E(D\theta_{qkv}D\theta_{qkv}^T) = E(\delta_{qkv}\delta_{qkv}^T \otimes x_{qkv}x_{qkv}^T) \\ &\approx E(\delta_{qkv}\delta_{qkv}^T) \otimes E(x_{qkv}x_{qkv}^T) = \Delta_{qkv} \otimes X_{qkv} \end{aligned} \qquad (9)$$

Also, $H_o = \Delta_o \otimes X_o$. Thus, the iHVP of the attention layer can be estimated as follows:

$$
\begin{aligned}
H_{att}^{-1} v_{att} &= \begin{bmatrix} \mathbf{H_{qkv}^{-1} v_{qkv}} \\ \mathbf{H_o^{-1} v_o} \end{bmatrix} = \begin{bmatrix} (\mathbf{\Delta_{qkv} \otimes X_{qkv})^{-1} v_{qkv}} \\ (\mathbf{\Delta_o \otimes X_o})^{-1} \mathbf{v_o} \end{bmatrix} \\
&= \begin{bmatrix} (\mathbf{\Delta_{qkv}^{-1} \otimes X_{qkv}^{-1}) v_{qkv}} \\ (\mathbf{\Delta_o^{-1} \otimes X_o^{-1}) v_o} \end{bmatrix} = \begin{bmatrix} \mathbf{vec(\Delta_{qkv}^{-1} V_{qkv} X_{qkv}^{-1})} \\ \mathbf{vec(\Delta_o^{-1} V_o X_o^{-1})} \end{bmatrix}
\end{aligned}
\tag{10}
$$

where $v_{att}, v_{qkv}, v_o$ represent the gradient of reference set $D_r$ on parameters $\theta_{att}, \theta_{qkv}, \theta_o$, respectively. Thus, the influence score of attention layers can be written as: $I_{\theta_{att}} = -\nabla L(\theta_{att}, z) H_{att}^{-1} v_{att}$.

To avoid the excessive memory usage of validation set gradients, we apply the Johnson-Lindenstrauss Lemma Johnson (1984) to reduce the dimensionality of both the iHVP computation results and the training data gradients $\nabla L(\theta, z)$. We achieve this by projecting $H_{att}^{-1} v_{att}$ onto a $d$-dimensional space, resulting in $Q^T H_{att}^{-1} v_{att}$. In the Lemma, each element of $Q$ is drawn from $\mathbb{R}^{P \times d}$, with $P$ representing the original dimensionality and $d$ denoting the reduced dimensionality.

## 4 EXPERIMENT

### 4.1 EXPERIMENT SETUP

**Dataset Preparation.** We use the entire 627B-token SlimPajama dataset (Soboleva et al., 2023) and 15000B-token FineWeb dataset (Penedo et al., 2024) as the candidate pool $D_c$. In the clustering process, the `BAAI/bge-large-en-v1.5 model` is employed to generate embeddings for the input data, and approximately 600 million data points from the candidate pool $D_c$ are clustered into 10,000 groups using the $k$-means algorithm. We use LAMBADA (Paperno et al., 2016), Openwebmath (Paster et al., 2023) and FLAN (Chung et al., 2024) as our reference (validation) set $D_r$, which are all widely used language modeling tasks and often serves as a validation benchmark for language model pretraining. (Yu et al., 2024; Xie et al., 2023; Hoffmann et al., 2022).

**Experimental Settings.** We train two transformer-based decoder-only language models, one with 1.3B parameters and the other with 7B parameters, both utilizing RoPE embeddings (Su et al., 2023) and a maximum context window of 2048 tokens (Touvron et al., 2023). Following the setting of MATES (Su et al., 2023), 30B tokens out of the 627B are selected for training using `Quad` and compare with baselines. The learning rate is set to $5 \times 10^{-5}$ for 1.3B model and $1 \times 10^{-5}$ for 7B model, the batch size is set to 4096, and the Adam optimizer is employed with hyperparameters $\beta_1 = 0.9, \beta_2 = 0.95, \epsilon = 10^{-8}$. As for Multi-Armed Badit, we set the $\alpha = 0.002$, sample proporation $\gamma = 0.05$ and the sample threshold $\tau$ as 0.0025.

**Baselines.** We compare our methods with several baselines. (1) `Random` samples data from the entire candidate dataset randomly. (2) `Qurating` uses the large language model to select data. (3) `DSIR` selects data instances that are similar to the LAMBADA, Openwebmath or FLAN dataset. (4) `PPL` uses perplexity-based data selection, *i.e.*, selecting data instances with the lowest perplexity scores. (5) `MATES` trains a surrogate model to evaluate the influence of each data instance on the target model.

**Evaluation Datasets.** To comprehensively evaluate the capabilities of pretrained models, we conduct experiments on various downstream tasks covering three significant categories:

General Knowledge: ARC-C, ARC-E (Clark et al., 2018), and SciQ (Welbl et al., 2017).

Commonsense Reasoning: HellaSwag (Zellers et al., 2019), SIQA (Sap et al., 2019), Wino-Grande (Sakaguchi et al., 2021), Logiqa (Liu et al., 2020).

Reading Comprehension: OpenbookQA (Mihaylov et al., 2018), and BoolQ (Clark et al., 2019).

Math: GSM8K (Cobbe et al., 2021), MATH (Hendrycks et al., 2021), OCW (Lewkowycz et al., 2022), SAT (Azerbayev et al., 2023), and MMLU STEM (Hendrycks et al., 2020)

Long-text Generation: WikiText (Merity et al., 2016), and HelloBench (Que et al., 2024).

Evaluations are conducted using the lm-evaluation-harness (Gao et al., 2023) framework and the average accuracy (*i.e.*, Overall Score) is reported for comparison.

Table 1: Overall Performance

| Selection Method | General Knowledge (3 tasks) | Commonsense Reasoning (4 tasks) | Reading Comprehension (2 tasks) | Overall | FLOPs |
|---|---|---|---|---|---|
| Random | 50.33 | 36.19 | 39.09 | 41.55 | 7.66 |
| DSIR | 50.37 $^{\uparrow 0.04}$ | 34.01 $^{\downarrow 2.18}$ | 38.80 $^{\downarrow 1.29}$ | 40.53 $^{\downarrow 1.02}$ | 7.66 |
| PPL | 48.71 $^{\downarrow 1.62}$ | **37.72** $^{\uparrow 1.53}$ | 38.57 $^{\downarrow 0.52}$ | 41.57 $^{\uparrow 0.02}$ | 9.51 |
| Semdedup | 50.99 $^{\uparrow 0.66}$ | 36.11 $^{\downarrow 0.08}$ | 39.44 $^{\uparrow 0.35}$ | 41.81 $^{\uparrow 0.26}$ | 8.11 |
| Qurating | 51.56 $^{\uparrow 1.23}$ | 35.93 $^{\downarrow 0.26}$ | 39.70 $^{\uparrow 0.61}$ | 42.01 $^{\uparrow 0.46}$ | 13.66 |
| MATES | 50.45 $^{\uparrow 0.12}$ | 36.06 $^{\downarrow 0.13}$ | 39.83 $^{\uparrow 0.74}$ | 41.93 $^{\uparrow 0.38}$ | 9.81 |
| Quad(ours) | **52.08** $^{\uparrow 1.75}$ | 37.03 $^{\uparrow 0.84}$ | **41.07** $^{\uparrow 1.98}$ | **42.94** $^{\uparrow 1.39}$ | 9.15 |

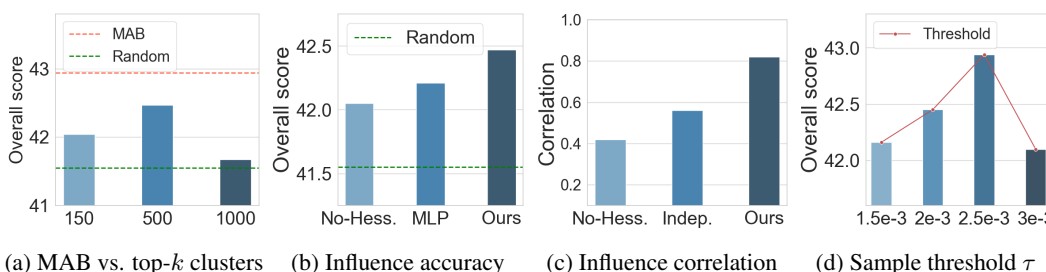

(a) MAB vs. top-$k$ clusters    (b) Influence accuracy    (c) Influence correlation    (d) Sample threshold $\tau$

Figure 4: (a) shows the effectiveness of the MAB method; (b) shows the accuracy of calculating the influence function on MLP and attention layers; (c) shows the correlation between Query, Key, Value layers impact a lot on the accuracy of influence calculation; (d) shows the model performance of varying sample threshold $\tau$.

## 4.2 RESULTS

**Overall Performance.** As demonstrated in Table 1, our method surpasses all the baseline methods in downstream tasks with zero-shot evaluation. To be specific, we can observe that on General Knowledge and Reading Comprehension tasks, Quad has the improvement of 1.75% and 1.98% respectively compared with Random. Quad outperforms DSIR and Semdedup because they use rule-based heuristics to select data without considering the model. Although PPL and MATES consider the model, they do not perform well because the former one always selects some simple and duplicated instances, and the surrogate model of the latter one is small and lacking of enough training data. Qurating generally performs the best among other baselines, but still worse than our approach, and it incorporates the highest FLOPs(1e19) because of the usage of LLMs for data selection. In terms of the FLOPs, we can observe that except the methods (*i.e.*, DSIR, SemDeDup) that use simple heuristics, we consume minimal computation resources because we sample from clusters without considering the entire candidate dataset like PPL, Qurating and MATES.

**Effectiveness of MAB.** This section evaluates the effectiveness of the MAB approach for data selection in contrast to the straightforward method of choosing the top-$k$ clusters with the highest influence scores for model training. To be specific, we randomly select an equivalent number of data points from the top 150, 500, and 1000 clusters. Figure 4a illustrates the trade-off between data quality and diversity: clusters with higher influence scores do not necessarily enhance model performance on downstream evaluation sets because of their lack of diversity. Hence, the multi-armed bandit method can more effectively capture the trade-off between quality and diversity across clusters, resulting in superior performance, as opposed to merely choosing the top-$k$ clusters.

**Effectiveness of Influence Calculation.** This experiment studies the effectiveness of our influence calculation method. In this section, we select the top 500 clusters with the highest scores using three methods: (1) no-Hessian (*i.e.*, computing the gradient similarity between training data and reference data (Pruthi et al., 2020)) without considering the Hessian matrix; (2) MLP(*i.e.*, calculat-

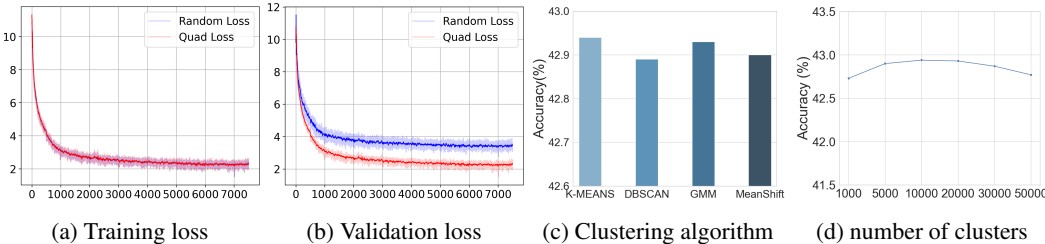

Figure 5: (a) training loss; (b) validation loss; (c) model performance on varying clustering algorithm; (d) model performance on varying number of cluster

ing influence function on MLP layers) and (3) `Ours` (*i.e.*, calculating influence function on both MLP and attention layers). From each cluster, we uniformly sample data to train the large model. As shown in Figure 4b, our solution (`MLP+Attention`) performs better than `MLP` because the attention layer considers more semantics. `no-Hessian` performs the worst because it does not precisely capture the impact of training data instances on the model without the Hessian matrix.

Also, we conduct experiments to verify the relationship between the Query, Key, Value matrices, which is shown in Figure 4c. We compare Pearson correlation coefficients of three methods with a baseline that computes the attention layer's influence score without acceleration. (1) `No-Hessian`(*i.e.*, computing the gradient similarity between training data and reference data) without considering the Hessian matrix; (2) `Independent` (*i.e.*, calculating the Hessian matrices of the query, key, and value layers independently) and (3) `Ours` (*i.e.*, calculating the Hessian matrices of the query, key, and value layers as a whole).

## 4.3 Continuous pretrain

In this experiment, we use Slimpajama (Soboleva et al., 2023) as the candidate dataset and Openwebmath (Paster et al., 2023) as the reference set on 7B model. For downstream tasks, we use GSM8K (Cobbe et al., 2021), MATH (Hendrycks et al., 2021), OCW (Lewkowycz et al., 2022), SAT (Azerbayev et al., 2023), and MMLU STEM (Hendrycks et al., 2020), calculating accuracy by averaging their scores. As shown in Figure 7d, selecting 400B random data instances performs worse than 100B selected by `Quad` because the larger set has irrelevant information, degrading the performance and `Quad` improves selection by identifying beneficial data for the specific domain.

## 4.4 Further experiment

**Generalizability of `Quad`** To demonstrate the generalizability and robustness of our method, we include FLAN (Chung et al., 2024) (a mix of multiple NLP tasks) as reference datasets . We use FLAN to train a transformer-based, decoder-only language model with 1.3B parameters on a new candidate dataset, FineWeb (Penedo et al., 2024). From this dataset, 100B tokens are selected for training using `Quad` alongside other baselines. As shown in the Figure 6(c), `Quad` exhibits superior accuracy compared to all baselines like DSIR, Semdedup, PPL, Qurating, and MATES. Notably, `Quad` shows a 0.97% accuracy improvement over Qurating, a leading baseline.

**Scalablity of `Quad`.** We increase the model size from 1.3B to 7B and choose 100B tokens from the Slimpajama dataset. As shown in the Figure 6(a), `Quad` outperforms other baselines in accuracy due to high-quality, diverse data selection. Additionally, `Quad` demonstrates good scalability compared to state-of-the-art baselines (Qurating and MATES) by using the K-FAC technique for faster influence computation and the MAB framework to quickly identify beneficial data instances.

**Effciency of `Quad`.** To highlight the importance of data selection, we compare `Quad` and direct training with clean data on 7B modes for downstream tasks. As shown in the Figure 7c, `Quad`-7-100B surpasses `Clean`-7B-100B because `Quad` selects data beneficial for the current training process, unlike the latter's random selection from SlimPajama. At the same time, `Quad`-7B-100B offers performance comparable to `Clean`-7B-400B, reducing computation by 75%.

## 4.5 Ablation Study

**Sampling Threshold of Influence ($\tau$).** Figure 4d illustrates that setting the threshold too high or low will both degrade the model performance. This is because the selected data instances tend to

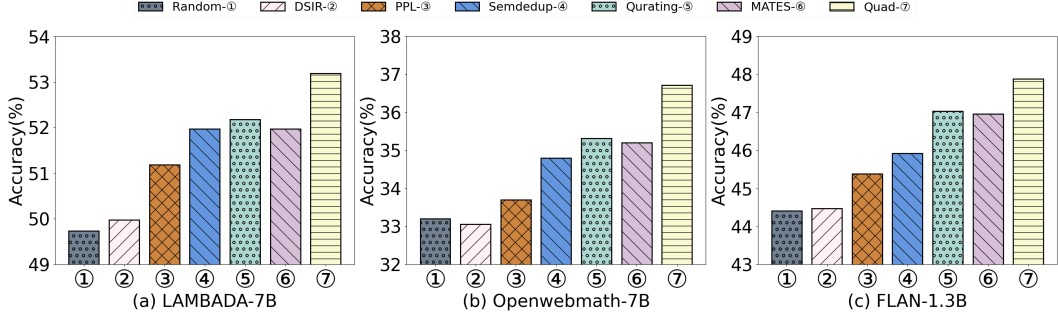

Figure 6: (a) 7B model use LAMBADA as reference set; (b) 7B model use Openwebmath as reference set; (c) 1.3B model use FLAN as reference set

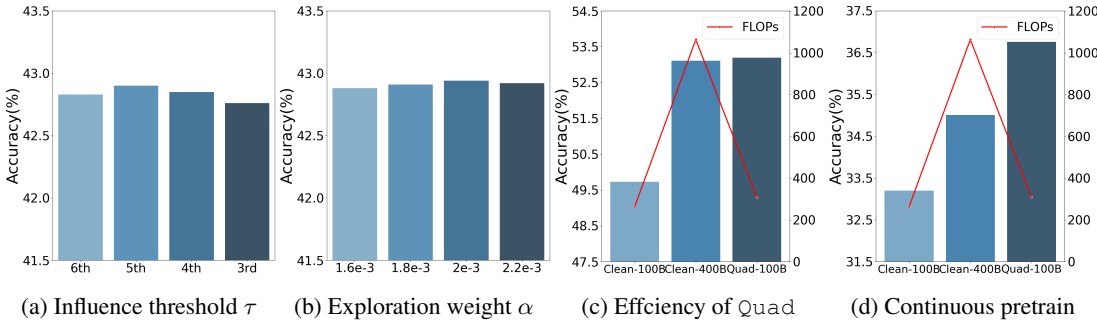

Figure 7: (a) model performance with different influence threshold $\tau$; (b) model performance with different $\alpha$; (c) the efficiency of Quad; (d) continuous pretrain

exist in few clusters with high influence scores, resulting in poor diversity. In contrast, when the threshold is set too low, the sampled instances will be from many clusters with low influence scores, which also degrades the model performance. Considering that the model's performance is highly sensitive to the influence threshold, a quick selecting method is proposed in Appendix F.

**Number of Clusters** We use the Elbow (Syakur et al., 2018) method to identify the optimal cluster number($k = 10000$) of the Slimpajama dataset. In Figure 5c, We plot the effect of different cluster numbers, which shows the model consistently performs well with around 10000 clusters. Too few clusters (i.e., $k = 1000$) cause high variance and poor representation, while too many clusters (i.e., $k = 500000$) result in redundancy and hinder diverse exploration, degrading performance.

**Clustering Algorithm** Moreover, we evaluate the performance of several typical clustering methods including GMM (Figueiredo & Jain, 2002), DBSCAN (Ester et al., 1996), and MeanShift (Cheng, 1995). The details of selecting optimal clustering parameters can be found in the Appendix D. As illustrates in Figure 5d, Quad is robust to clustering algorithms on downstream tasks.

$\alpha$ **for Quality-Diversity Balance.** Our approach employs $\alpha$ to balance the diversity and quality in the MAB framework. As shown in figure 7b, when $\alpha$ is small, the MAB framework prioritizes high-influence clusters and risks local optima due to reduced diversity. Conversely, when $\alpha$ is large, it overemphasizes diversity at the expense of quality, limiting model performance gains.

## 5 CONCLUSION & LIMITATIONS

This paper presents Quad, a method to balance diversity and quality in pretraining data selection. Quad employs influence functions to identify beneficial data by clustering and selecting representative subsets. Each cluster is treated as an arm in an MAB framework due to uncertain influence scores, allowing sampling from quality clusters to estimate influence scores accurately while maintaining diversity. We also adapt the influence function for attention layers and enhance calculation efficiency for better data impact assessment. As an exploratory research work, the ability of Quad to generalize to larger model sizes needs to be further explored, which is a challenging and significant issue for the method to be used in practice.

ACKNOWLEDGMENTS

Ju Fan is supported by the NSFC (62436010 and 62441230), the Beijing Natural Science Foundation (L244010 and L222006), and the Research Funds of Renmin University of China. Chengliang Chai is supported by National Key Research and Development Program of China (2024YFC3308200), NSFC (62472031), CCF-Baidu Open Fund (CCF-Baidu202402) and Huawei. Ye Yuan is supported by Beijing Natural Science Foundation (L241010), the National Key R&D Program of China (Grant No.2022YFB2702100), and the NSFC (Grant Nos. 61932004, 62225203, U21A20516). Guoren Wang is supported by the NSFC (62427808, U2001211), the Liaoning Revitalization Talents Program (Grant No.XLYC2204005).

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

# A    DETIALED RESULT

Table 2: Performance Comparison

| Selection Method | General Knowledge | | | | Commonsense Reasoning | | | | | Reading Comprehension | | | Overall |
|---|---|---|---|---|---|---|---|---|---|---|---|---|---|
| | arc-e | arc-c | sciq | **avg** | logiqa | hellaswag | siqa | winogrande | **avg** | openbookqa | boolq | **avg** | |
| Random | 50.27 | 20.31 | 80.40 | **50.33** | 21.20 | 34.11 | 38.49 | 50.99 | **36.20** | 17.60 | 60.58 | **39.09** | **41.55** |
| Semdedup | 51.35 | 20.73 | 80.90 | **50.99** | 19.05 | 34.56 | 39.30 | 51.54 | **36.11** | 18.80 | 60.09 | **39.45** | **41.81** |
| MATES | 50.00 | 21.25 | 80.10 | **50.45** | 21.66 | 33.90 | 38.69 | 52.17 | **36.61** | 19.00 | 60.67 | **39.84** | **41.94** |
| PPL | 45.41 | 20.82 | 79.90 | **48.71** | 20.43 | 35.92 | 39.92 | 54.62 | **37.72** | 18.80 | 58.35 | **38.58** | **41.57** |
| DSIR | 49.28 | 20.14 | 81.70 | **50.37** | 21.20 | 30.89 | 35.98 | 47.99 | **34.02** | 16.20 | 61.41 | **38.81** | **40.53** |
| Qurating | 52.10 | 23.29 | 79.80 | **51.56** | 20.43 | 33.57 | 39.05 | 50.67 | **35.93** | 18.00 | 61.41 | **39.71** | **42.04** |
| Quad(ours) | 52.27 | 21.76 | 82.20 | **52.08** | 22.89 | 34.41 | 38.74 | 52.09 | **37.03** | 20.00 | 62.14 | **41.07** | **42.94** |
| top$k$-cluster | | | | | | | | | | | | | |
| top-150 | 48.61 | 20.90 | 79.00 | **49.50** | 23.66 | 34.51 | 39.00 | 51.78 | **37.24** | 19.20 | 61.74 | **40.47** | **42.04** |
| top-500 | 51.05 | 21.25 | 79.70 | **50.67** | 22.73 | 34.40 | 39.20 | 52.41 | **37.19** | 18.80 | 62.76 | **40.78** | **42.48** |
| top-1000 | 49.96 | 20.99 | 80.40 | **50.45** | 21.97 | 34.00 | 38.74 | 50.2 | **36.23** | 18.20 | 60.61 | **39.41** | **41.67** |

Table 3: Ablation Study of Threshold $\tau$

| Threshold | General Knowledge | | | | Commonsense Reasoning | | | | | Reading Comprehension | | | Overall |
|---|---|---|---|---|---|---|---|---|---|---|---|---|---|
| | arc-e | arc-c | sciq | **avg** | logiqa | hellaswag | siqa | winogrande | **avg** | openbookqa | boolq | **avg** | |
| 0.0015 | 51.26 | 21.16 | 80.20 | **50.87** | 21.51 | 33.92 | 39.00 | 51.07 | **36.38** | 19.60 | 61.74 | **40.67** | **42.16** |
| 0.0020 | 52.23 | 22.27 | 80.70 | **51.73** | 22.89 | 34.77 | 38.33 | 50.20 | **36.55** | 19.20 | 61.50 | **40.35** | **42.45** |
| 0.0025 | 52.27 | 21.76 | 82.20 | **52.08** | 22.89 | 34.41 | 38.74 | 52.09 | **37.03** | 20.00 | 62.14 | **41.07** | **42.94** |
| 0.0030 | 50.25 | 19.62 | 80.80 | **50.22** | 22.27 | 33.96 | 38.96 | 53.28 | **37.12** | 20.60 | 59.20 | **39.90** | **42.10** |

Table 4: Effectiveness of Influence Calculation

| Method | General Knowledge | | | | Commonsense Reasoning | | | | | Reading Comprehension | | | Overall |
|---|---|---|---|---|---|---|---|---|---|---|---|---|---|
| | arc-e | arc-c | sciq | **avg** | logiqa | hellaswag | siqa | winogrande | **avg** | openbookqa | boolq | **avg** | |
| Random | 50.27 | 20.31 | 80.40 | **50.33** | 21.20 | 34.11 | 38.49 | 50.99 | **36.20** | 17.60 | 60.58 | **39.09** | **41.55** |
| No-Hessian | 49.03 | 20.99 | 80.50 | **50.20** | 22.58 | 33.40 | 38.89 | 52.41 | **36.82** | 19.20 | 61.50 | **40.35** | **42.06** |
| MLP | 50.63 | 21.50 | 78.90 | **50.34** | 22.89 | 33.32 | 38.74 | 52.57 | **36.88** | 19.60 | 61.77 | **40.69** | **42.21** |
| Ours | 51.05 | 21.25 | 79.70 | **50.67** | 22.73 | 34.40 | 39.20 | 52.41 | **37.19** | 18.80 | 62.76 | **40.78** | **42.48** |

Table 5: Model Architecture

| Hyperparameter | Value |
|---|---|
| Vocabulary Size | 32,000 |
| MLP Ratio | 8/3 |
| Hidden Dimension Size | 2048 |
| Number of Layers | 24 |
| Number of Attention Heads | 16 |
| Number of KV Attention Heads | 16 |
| RoPE Base | 10,000 |
| Maximum Context Window Length | 1024 |
| Number of Parameters | 1,345,423,360(1.3B) |

# B    EFFECTIVE SOLUTION OF SUITABLE SAMPLE RATIO

For this hyperparameter, we explore the extreme cases in the 100B scenario.

[*Extreme cases.*] The extreme cases of $\gamma$ are very small and large sampling ratios. As shown in the table below, a very small sampling ratio (*i.e.*, $\gamma = 0.1\%$) leads to low model accuracy (2.77% lower than the accuracy of the most appropriate sampling ratio, *i.e.*, 5%) due to the inaccurate estimates of influence scores. We can observe that a large sampling ratio (*i.e.*, $\gamma = 20\%$) does not improve the model performance much because a relatively accurate influence estimation is sufficient, but incorporates high computational costs.

[*Metric and Criteria.*] At a high level, we compute the most appropriate sampling ratio by sampling several clusters. For each sampled cluster, we compute the influence scores of its data instances and calculate the true average score. Then, we use the difference between the estimated score and the true score as the metric to select the best sampling ratio $\tau$. The high-level idea is to compute the true average influence scores of several sampled clusters and then identify the appropriate sampling ratio to accurately approximate the average.

[*Specific hypermarameter selection strategy.*] Specifically, after clustering the data instances in the candidate dataset, we randomly sample several clusters. Initially, we sample 1% data instances from the cluster and compute the average score. if the difference is within 10%, we set this proportion (1%) as the suitable sampling ratio for that cluster. Otherwise, we increase the ratio by 1%, to 2% , compute a new average, and repeat this process until the difference is within 10%. Finally, we average the suitable ratios from the sampled clusters to find the overall appropriate sampling ratio.

In this way, although we sample once, there would be a fairly accurate estimation of the average influence score. Since we are likely to sample multiple times, the estimation will be even more accurate. Besides, as data instances within each cluster exhibit similarity, by sampling a small fraction, we can have an estimation that is precise enough to select high-quality data for good model performance, thereby keeping computational costs low.

In our experiments, based on the Slimpajama dataset, we have determined that the optimal ratio is about 5%, as indicated in the paper. Here, we add another ablation study to demonstrate the effectiveness of this strategy. Specifically, we increase the sampling ratio from 1%, run the `Quad` algorithm respectively, and report the model accuracy of training with the selected data using the sampling ratio. We can observe from the below table that with the ratio increasing from 1%, the model accuracy increases because the influence estimation is more accurate. When the ratio exceeds 5%, the accuracy remains stable because that ratio is large enough to have an accurate estimation. Hence, it is not necessary to keep increasing the ratio, which will consume more computational costs.

Table 6: Ablation Study of Sampling Ratio

| Sampling Ratio | 0.1% | 1 % | 2 % | 3 % | 4 % | 5 % | 6 % | 7 % | 8 % | 9 % | 10 % | 20 % |
|---|---|---|---|---|---|---|---|---|---|---|---|---|
| Accuracy | 44.97 | 46.11 | 46.92 | 47.36 | 47.63 | 47.88 | 47.93 | 47.96 | 47.97 | 47.97 | 47.97 | 47.99 |

## C  STATISTICAL SIGNIFICANCE TESTS

We run the experiment six times and report the statistics significance tests. The results are as follows:

Table 7: Statistic Significance of Model Performance

|  | Exp-1 | Exp-2 | Exp-3 | Exp-4 | Exp-5 | Exp-6 | Avg $\pm$ std |
|---|---|---|---|---|---|---|---|
| Random | 41.52% | 41.32% | 41.68% | 41.72% | 41.57% | 41.55% | 41.56%$\pm$0.13% |
| DSIR | 40.59% | 40.47% | 40.72% | 40.29% | 40.37% | 40.53% | 40.49%$\pm$0.14% |
| PPL | 41.35% | 41.72% | 41.55% | 41.60% | 41.61% | 41.57% | 41.57%$\pm$0.11% |
| Semdedup | 41.60% | 41.94% | 41.87% | 42.04% | 41.91% | 41.81% | 41.86%$\pm$0.14% |
| Qurating | 42.03% | 41.89% | 41.81% | 42.21% | 42.13% | 42.01% | 42.01%$\pm$0.13% |
| MATES | 42.10% | 41.76% | 42.20% | 41.70% | 41.81% | 41.93% | 41.92%$\pm$0.18% |
| Quad(ours) | 43.11% | 42.91% | 43.03% | 43.02% | 42.87% | 42.94% | 42.98%$\pm$0.08% |

Based on the above results, we perform a significance test utilizing t-tests. We make the following two hypothesis. (1) Null hypothesis ($H_0$): The average accuracy of `Quad` does not exceed the average of baselines; and (2) Alternative hypothesis ($H_1$): The average performance of `Quad` exceeds that of the baselines. We set the confidence level as 99%, with a significance level of $\alpha = 0.01$. The p-value represents the probability of observing the test statistic under the assumption that $H_0$ is true. A small p-value indicates a lower likelihood of observing the current result if $H_0$ holds.

For each baseline, we can observe $p < \alpha$, and thus we reject $H_0$ and accept $H_1$, indicating that the average performance of `Quad` is greater than that of baselines. Since the confidence level of

Table 8: T Test of Different Selection Methods versus Quad

| t_test | Random | DSIR | PPL | Semdedup | Qurating | MATES |
|--------|--------|------|-----|----------|----------|-------|
| p-value | 7e-10 | 5e-12 | 2e-10 | 1e-08 | 4e-08 | 2e-07 |

t-test surpasses 99%, we can conclude that the superiority of `Quad` compared to other baselines is statistically significant.

## D   ABLATION STUDY OF CLUSTERING NUMBERS AND ALGORITHMS

In this part, we conduct experiments to explore extreme cases of selecting 100B data.

[*Extreme cases.*] As shown in the table below, a very small number of clusters (i.e., $k = 100$) leads to poor accuracy (2.77% lower than the accuracy of the best cluster number) due to the high variance of influence scores in each cluster. Thus, the sampled instances cannot well represent the cluster. Similarly, when the cluster number is too high (i.e., $k = 1,000,000$), there will be many clusters that are in fact contain similar data instances, and thus it is relatively hard to explore diverse clusters, thereby leading to the performance degradation (1.37% lower than the accuracy of the best cluster number).

[*Metric and Criteria.*] We use the metric Within-Cluster Sum of Squares (WCSS) to select the best cluster number using the well-known Elbow [1] algorithm. WCSS is the sum of squared distances between each data instance and its cluster center, i.e., WCSS=$\sum_{i=1}^{k} \sum_{x \in C_i} \|x - \mu_i\|$. At a high level, the criteria should be that within each cluster, data instances are close to each other, based on which it is better for different cluster centers to be far away from each other. Based on the criteria, the Elbow algorithm leverages the WCSS as a measurement to iteratively select an appropriate cluster number, as follows.

[*Specific hypermarameter selection strategy.*] To be specific, Elbow begins with a small $k$, and with $k$ increasing, WCSS first decreases rapidly and then slows down. Then, we identify the "elbow point" where the decreasing rate becomes slow as the best $k$. Thus, within each cluster, data points are sufficiently close to one another. Furthermore, given that $k$ remains modest, different cluster centers tend to maintain a distance from each other. From the following table we can observe that the model consistently performs well when the cluster number is close to 10,000.

Table 9: Ablation Study of Cluster Numbers

| Cluster Numbers | 100 | 1,000 | 5,000 | 10,000 | 20,000 | 50,000 | 100,000 | 1,000,000 |
|-----------------|-----|-------|-------|--------|--------|--------|---------|-----------|
| Accuracy | 45.11 % | 46.93% | 47.21% | 47.88% | 47.73% | 47.73% | 47.06% | 46.51% |

*Clustering algorithms.* In terms of the clustering algorithms, we also added experiments to show that `Quad` is not sensitive to clustering algorithms mainly because different algorithms have their own strategies to select appropriate parameters, which follows the criteria mentioned above. Under the criteria, in general, `Quad` can perform well by considering both the quality and diversity.

Specifically, we evaluate the performance of several typical clustering methods including GMM (Figueiredo & Jain, 2002), DBSCAN (Ester et al., 1996) and MeanShift (Cheng, 1995). Considering that the clustering results are affected by the parameters of clustering algorithms, we use different methods to select proper parameters. For GMM, we can use the AIC score (Aho et al., 2014) to determine the appropriate number of components. For DBSCAN, there are 2 key parameters: (1) $eps$(the radius of a neighborhood w.r.t. some data points) and (2) $minPts$ (a data point is considered as a core point if at least $minPts$ data points are within $eps$ of it). They can be set using the method in (Schubert et al., 2017). Mean-Shift is a centroid-based method that updates the centroids to be the mean of the points within a given region. The size of the region is controlled by $bandwidth$, which can be set by the estimation of the bandwidth. In this set of experiments, the experimental settings of pretraining 1.3B model are consistent with those reported in our paper. The above result shows that `Quad` is not sensitive to the clustering algorithms.

Table 10: Ablation Study of Clustering Methods

| Clustering Methods | K-MEANS | DBSCAN | GMM | MeanShift |
|---|---|---|---|---|
| Accuracy | 47.88 % | 47.79 % | 47.69 % | 47.71 % |

## E  FLOPs CALCULATION

FLOPs is the number of floating point operations performed by GPUs. Many state-of-the-art methods [1,2,3] use it to measure the consumption of GPU computing resources. In our experiments, FLOPs is collected directly in the data selection process using the Python code:

```python
import torch
import torch.nn as nn
from torch.profiler import profile, ProfilerActivity

model = nn.Linear(1024, 512).cuda()
input_data = torch.randn(128, 1024).cuda()
with                    profile(activities=[ProfilerActivity.CPU,
ProfilerActivity.CUDA],
    with_flops=True) as prof:
        model(input_data)
print(prof.key_averages().table(sort_by="flops", row_limit=10))
```

## F  EFFECTIVE SOLUTION OF INFLUENCE THRESHOLD

Recap that the sampling threshold of influence ($\tau$) is utilized to determine whether data instances sampled from each cluster should be fed into LLMs for training. For this hyperparameter, we also explore the extreme cases in the 100B scenario.

[*Extreme cases.*] If $\tau$ is very large (*e.g.*, 2.92e-3, close to the smallest influence score in the candidate dataset), only data instances with high influence scores are selected. This typically results in reduced diversity among instances, which in turn can negatively impact the model performance. If $\tau$ is small enough (*e.g.*, 1.76e-3, close to the largest influence score in the candidate dataset), many low-quality data instances with low influence scores are selected. This also hurts the model performance.

Table 11: Ablation Study of Influence Threshold

| Influence threshold($\tau$) | Bucket | Accuracy |
|---|---|---|
| 2.92e-3 | 1-th | 46.87% |
| 2.60e-3 | 3-th | 47.30% |
| 2.47e-3 | 4-th | 47.88% |
| 2.38e-3 | 5-th | 47.65% |
| 2.17e-3 | 6-th | 47.19% |
| 1.76e-3 | 10-th | 45.26% |

[*Metric and Criteria.*] We use the model performance on the validation (reference) set as the evaluation metric. The criteria is that we sample some data instances from the candidate dataset. Based on the samples, we try different thresholds, evaluate on the validation test and select the best threshold.

[*Specific hypermarameter selection strategy.*] The strategy of selecting an appropriate $\tau$ consists of the following steps. (1) Considering the efficiency issue, we sample about 20% data instances to form a new candidate dataset from the original one to tune the parameter. (2) We cluster over the candidate dataset, sample some instances from each cluster and compute their influence scores, which are utilized to capture the distribution of influence scores. (3) We rank these scores in descending order and assign them to 20 buckets, so as to derive 20 thresholds, among which we select the best one. For example, the first threshold corresponds to the highest influence score among the top 5% instances (the first bucket). The last threshold is the lowest influence score among all in-

stances of buckets. (4) Finally, we select data instances via the `Quad` algorithm using the above thresholds, evaluate on the validation set, and select the best one.

## G  SCALABILITY OF QUAD

In this response, we first discuss the advantage of using `Quad` to select data instances. We then run experiments to validate the argument.

Overall, we agree with the reviewer that data cleaning is likely to enhance the model performance. However, the obtained data after cleaning is still large-scale, which is prohibitively expensive to train, but not always necessary. This is because a subset of judiciously selected data can achieve competitive or even superior model performance compared with training with all clean data. The reasons are two-fold. First, `Quad` is a model-aware data selection method, which uses the data influence to measure the impact of each data instance on the current training process, but coarse-grained data cleaning methods are not customized to the target tasks. Therefore, given a user-specific task, simply using all available data to train may not the optimal choice. Second, `Quad` also considers the diversity for data selection, and thus the selected small part of data instances that effective for the current pretraining process.

Overall, the model-aware data selection is of great significance because (1) it is customized to the reference set, which can much improve the model performance with a small number of data instances; and (2) the model can be trained over these selected small amounts of data, thus greatly improving the training efficiency.

Next, we run two new experiments to validate this.

First, we would like to clarify that the SlimPajama dataset used in our experiment has already undergone multiple rounds of coarse-grained data cleaning processes Soboleva et al. (2023), including short documents filtering, deduplication, minhash generation, etc. Therefore, to demonstrate the significance of the data selection method, we compare the performance of `Quad` and directly training with clean data with different data sizes with 1.3B and 7B models, evaluating on the long-text generation tasks mentioned in the last response. As shown in the below table, we can observe that `Quad`-1.3B-100B outperforms `Clean`-1.3B-100B because `Quad` judiciously selects data that benefits the current training process while the latter one just selects 100B data randomly from the clean SlimPajama dataset without considering the current model state. Because of the similar reason, `Quad`-1.3B-100B achieves comparable model performance with `Clean`-1.3B-400B, which saves 75% of computation costs. We can also observe that training on 7B model has a similar result.

Table 12: Model Performance and Flops with Different Method and Training Tokens

| Methods | Accuracy | FLOPs |
|---|---|---|
| Clean-1.3B-100B | 44.41 % | 58.1 |
| Clean-1.3B-400B | 47.66% | 232.5 |
| Quad-1.3B-100B | 47.88 % | 72.3 |
| Clean-7B-100B | 49.73% | 263.1 |
| Clean-7B-400B | 53.11% | 1063.7 |
| Quad-7B-100B | 53.19% | 307.6 |

Secondly, we add another dataset Openwebmath (Paster et al., 2023) as our new reference dataset. It is based on the Slimpajama dataset, which aims at selecting data to improve the mathematical skill of the model, i.e., a specific domain. We use GSM8K (Cobbe et al., 2021), MATH (Hendrycks et al., 2021), OCW (Lewkowycz et al., 2022), SAT (Azerbayev et al., 2023), MMLU STEM (Hendrycks et al., 2020) as the downstream tasks and the final accuracy is computed the average score of these downstream tasks. As shown in Table 12, we can observe that random selection of 400B data instances does not perform better than selecting 100B by `Quad` on downstream mathematical evaluation datasets. This is because the 400B data instances contain a significant amount of information irrelevant to mathematics, which degrades the model performance, while `Quad` is a model-aware method that accurately identifies the data that benefits the specific domain($i.e.$, the math domain).

Table 13: Model Performance in Math Evaluation

| Methods | Accuracy |
|---|---|
| Clean-7B-100B | 33.20 % |
| Clean-7B-400B | 35.01 % |
| Quad-7B-100B | 36.76 % |

Moreover, we train a 7B model and evaluate all methods on the SlimPajama dataset (Soboleva et al., 2023), using LAMBADA as the reference dataset. In this experiment, 100B tokens out of 627B are selected for training using Quad. As shown in the table below, Quad still demonstrates superior model performance on 7B model compared to all baselines, such as the DSIR, Semdedup, PPL, Qurating and MATES.

Table 14: Model Performance with a Larger Model Size

| Methods | Accuracy | FLOPs |
|---|---|---|
| Random-100B | 49.73% | 263.1 |
| Random-400B | 53.11% | 1063.7 |
| DSIR-100B | 49.97% | 263.1 |
| PPL-100B | 51.18% | 325.3 |
| Semdedup-100B | 51.97% | 275.7 |
| Qurating-100B | 52.18% | 375.6 |
| MATES-100B | 51.97% | 313.3 |
| Quad-100B | 53.19% | 307.6 |

Note that our CS function is to balance the exploration (i.e., diversity) and exploitation (i.e., quality) inspired by the upper confidence bound score (Auer, 2002), which is a typical solution of MAB. However, certainly our approach can use other CS functions. Here we discuss two intuitive alternatives, i.e., the exploitation-only and exploration-only methods to demonstrate the superiority of us. The former one samples some data instances from each cluster, computes their influence scores and selects instances from clusters with high influence scores. The latter one just randomly samples from different clusters without considering the data quality. As shown in the table below, we can observe that for both 1B and 7B models, our method outperforms the two baselines because we consider both the data quality and diversity to select data instances.

## H CS FUNCTION

Note that our CS function is to balance the exploration (i.e., diversity) and exploitation (i.e., quality) inspired by the upper confidence bound score (Auer, 2002), which is a typical solution of MAB. However, certainly our approach can use other CS functions. Here we discuss two intuitive alternatives, i.e., the exploitation-only and exploration-only methods to demonstrate the superiority of us. The former one samples some data instances from each cluster, computes their influence scores and selects instances from clusters with high influence scores. The latter one just randomly samples from different clusters without considering the data quality. As shown in the table below, we can observe that for both 1B and 7B models, our method outperforms the two baselines because we consider both the data quality and diversity to select data instances.

Table 15: Model Performance with Different Size and Sampling Method

| Model | Quad Score | exploitation-only | exploration-only |
|---|---|---|---|
| 1B | 42.94 % | 42.36% | 41.60 % |
| 7B | 53.19 % | 52.56 % | 49.33% |

## I    GENERALIZABILITY

To show the generalizability and robustness of our method, we add FLAN (Chung et al., 2024) (the mixture of multiple NLP tasks) and Openwebmath  (Paster et al., 2023) (a math-related dataset) as reference datasets, associated with 7 new downstream tasks. We also add one candidate dataset and a 7B downstream model.

Specifically, we first use FLAN as a new reference dataset to train a 1.3B model on a new candidate dataset (i.e., FineWeb  (Penedo et al., 2024), approximately 15000B tokens). In this experiment, 100B tokens out of 15000B tokens are selected for training using `Quad` and other baselines. As shown in the table below, `Quad` still demonstrates superior accuracy compared to all baselines, such as the DSIR, Semdedup, PPL, Qurating and MATES. For example, we can observe that `Quad` has an improvement of 0.97% on model accuracy compared with the Qurating, which is a state-of-the-art baseline. In addition, we add two downstream tasks (i.e., WikiText (Merity et al., 2016), HelloBench (Que et al., 2024) , and for WikiText, the lower the score, the better the model performance) about long-text generation tasks. As shown in the table below, on the candidate dataset Slimpajama, for the 7B model, we can observe that `Quad` still outperforms other baselines because we simultaneously consider the quality and diversity in data selection.

Table 16: Model Performance with Reference Set FLAN

| Methods | Random | DSIR | Semdedup | PPL | Qurating | MATES | Quad |
|---------|--------|------|----------|-----|----------|-------|------|
| Accuracy | 44.41% | 44.47% | 45.38% | 45.92% | 47.03 % | 46.96 % | 47.88% |

Table 17: Model Performance on Long-Text Generation Tasks

| | Random | Qurating | MATES | Quad |
|---|--------|----------|-------|------|
| WikiText | 16.73 | 15.36 | 15.51 | 14.81 |
| HelloBench | 3.81 | 5.37 | 4.39 | 6.33 |

Moreover, we also add the Openwebmath dataset as another new reference dataset to train a 7B model on the Slimpajama dataset, which aims at selecting data to improve the mathematical skill of the model. Correspondingly, we also includeGSM8K (Cobbe et al., 2021), MATH (Hendrycks et al., 2021), OCW (Lewkowycz et al., 2022), SAT (Azerbayev et al., 2023), MMLU STEM (Hendrycks et al., 2020) as new downstream tasks to evaluate the model's mathematical capabilities, with the final accuracy calculated as the average score across these tasks. As shown in the table below, `Quad` still demonstrates superior model performance compared to all baselines.

Table 18: Model Average Performance on 5 Math Tasks

| Methods | Random | DSIR | Semdedup | PPL | Qurating | MATES | Quad |
|---------|--------|------|----------|-----|----------|-------|------|
| Accuracy | 33.20% | 33.05% | 33.70% | 34.79% | 35.31% | 35.20% | 36.71% |

This experiment shows that our method can generalize well to various types of reference datasets and downstream tasks.

## J    RELIABILITY OF OUR EXPERIMENTAL RESULT

For our baselines Qurating & MATES, the performance improvement comparing with the best baseline in their papers are 1.9% and 0.6% respectively. In our work, `Quad` surpasses the best baseline (i.e., Qurating) by 0.93%, so the improvement is not marginal. Specifically, although `MATES` also selects data considering the current training process, it does not perform well because the surrogate model is not accurate enough due to lacking of enough training data. `Qurating` generally performs the best among other baselines, but still worse than our approach because it does not consider the varience of model state during the training process, and it incorporates the highest FLOPs (1e19)

because of the usage of LLMs for data selection. In terms of the FLOPs, we can observe that we consume minimal computational resources because our MAB solution samples from clusters without iterating the entire candidate dataset like `Qurating` and `MATES`.

For DSIR, we would like to clarify that in the literature, i.e., the papers of our baselines, DSIR also performs worse than random selection. This is consistent with our results. For example, in Qurating, random selection achieves a 2% higher accuracy than DSIR. In MATES, DSIR also does not perform better than random selection. The reason is that DSIR selects data instances whose n-gram features are similar to the instances in the validation set, which cannot capture the data semantics and the downstream model performance, leading to a poor generalization ability. In addition, purely relying on the similarity of the strings to select data will incorporate many duplicated instances, which may even hurt the model performance.

To better demonstrate the benefits of our method `Quad`, we enlarge the model size from 1.3B to 7B, using LAMBADA as the reference dataset to select 100B tokens from Slimpajama dataset . As demonstrated in the following table, `Quad` outperforms other baselines in terms of accuracy and has good scalability (achieving low FLOPs).

Table 19: Model Performance with Larger Model Size

| Methods | Random | DSIR | Qurating | MATES | Quad |
|---|---|---|---|---|---|
| Accuracy | 49.73% | 49.97% | 52.18% | 51.97% | 53.19% |

For the training loss, we find that `Quad` converges successfully at a rate similar to the random selection. It is notable that for the validation loss, `Quad` converges faster than random selection because we can discover data instances that are beneficial for the current training process.

