# OpenReview forum: "Harnessing Diversity for Important Data Selection in Pretraining Large Language Models"
_ICLR.cc/2025/Conference — ICLR 2025 Spotlight_

### Official Review · Reviewer_Q54u · 2024-10-18

**Soundness:** 2
**Presentation:** 2
**Contribution:** 3
**Rating:** 6
**Confidence:** 3

**Summary:**

This work proposes to balance the diversity and quality in data selection for LLMs pre-training. The author adopts the MAB method to dynamically estimate and sample data points from various clusters. Additionally, they introduce a novel influential calculation approach to estimate the data point more accurately while improving processing efficiency. Their experimental results on extensive benchmarks validate the effectiveness and efficiency of the proposed method.

**Strengths:**

* The introduction of MAB to LLMs' pre-training data selection is reasonable and interesting, addressing a practical problem about diversity and efficiency.
* The experiment results demonstrate that their proposed method significantly improves the pre-trained LLM's performance while maintaining efficiency.

**Weaknesses:**

* The experiments are not thorough. The experiment and analysis only take up 2 pages, which fails to thoroughly discuss many important components and designs of the proposed method. For example, whether the CS function could be designed in another way? Also, while the evaluation is conducted in various benchmarks, the authors only adopt experiments in a single LLM. This also makes people doubt the generalization of their method. I suggest the author to fully utilize the 10-page space and conduct further experiments.

* According to ablation study results, the model's performance is quite sensitive to the sampling threshold of influence. And there are no hyper-parameter selection strategies introduced in the paper. All these make me worry about the robustness of the proposed method to apply in real-world scenarios.

**Questions:**

* How do you choose the sampling threshold hyper-parameter in the main experiment? Have you tuned it in a validation set?

---

> ### Author Response · Authors · 2024-11-23
>
> $\textbf{Overall response:}$
>
> Thanks for your valuable comments! Based on your comments, we have added a 7B model, 2 reference datasets, 7 downstream tasks  and in total 12 additional experiments to fully utilize the 10 pages and put some experiments in the Appendix, so as to demonstrate the generalizability and robustness of $\texttt{Quad}$.  We also clarify your concerns about the sensitivity of the sampling threshold of influence in our paper. We promise to include the above discussions into the final version if the paper is accepted.
>
> $\textbf{Weakness 1:}$ The experiments are not thorough. The experiment and analysis only take up 2 pages, which fails to thoroughly discuss many important components and designs of the proposed method. For example, whether the CS function could be designed in another way? Also, while the evaluation is conducted in various benchmarks, the authors only adopt experiments in a single LLM. This also makes people doubt the generalization of their method. I suggest the author to fully utilize the 10-page space and conduct further experiments.
>
> $\textbf{Response for Weakness 1:}$
> Thanks for the valuable suggestion to improve our paper! As the reviewer suggested, to demonstrate the robustness and effectiveness of $\texttt{Quad}$,  we have run new experiments to evaluate  different model sizes, different downstream tasks, different candidate datasets and run various ablation studies (including the clustering algorithms, clustering numbers, sampling influence threshold, etc.).
> We have included the above discussion into Appendix G and H of the revised paper.
>
> Specifically, we train a 7B model from scratch and evaluate all methods on the SlimPajama dataset, using LAMBADA as the reference dataset. In this experiment, 100B tokens out of  627B are selected for training using $\texttt{Quad}$.
> As shown in the table below, $\texttt{Quad}$ still demonstrates superior model performance on 7B model compared to all baselines, such as the DSIR, Semdedup, PPL, Qurating and MATES.
> Also, $\texttt{Quad}$ achieves similar performance with the random selection which trains the model with 400B tokens, consuming 307.6 FLOPs instead of 1063.7 FLOPs (1e19) for 400B tokens.
> However, $\texttt{Quad}$ only uses 100B tokens, thus greatly improving the training efficiency.
>
> | Methods| Accuracy     | FLOPs   |
> |-|-|-|
> | $\texttt{Random}$-100B | 49.73    | 263.1   |
> | $\texttt{Random}$-400B | 53.11    | 1063.7  |
> | $\texttt{DSIR}$-100B   | 49.97    | 263.1   |
> | $\texttt{PPL}$-100B    | 51.18    | 325.3   |
> | $\texttt{Semdedup}$-100B | 51.97  | 275.7   |
> | $\texttt{Qurating}$-100B | 52.18  | 375.6   |
> | $\texttt{MATES}$-100B  | 51.97    | 313.3   |
> | $\texttt{Quad}$-100B   | 53.19    | 307.6   |
>
>
> Note that our CS function is to balance the exploration (i.e., diversity) and exploitation (i.e., quality) inspired by the upper confidence bound score[1], which is a typical solution of MAB.
> However, certainly our approach can use other CS functions. Here we discuss two intuitive alternatives, i.e., the exploitation-only and exploration-only methods to demonstrate the superiority of us. The former one samples some data instances from each cluster, computes their influence scores and selects instances from clusters with high influence scores. The latter one just randomly samples from different clusters without considering the data quality. As shown in the table below, we can observe that for both  1B and 7B models, our method outperforms the two baselines because we consider both the data quality and diversity to select data instances.
>
> | Model   | $\texttt{Quad}$ Score | Exploitation-only | Exploration-only |
> |-|-|-|-|
> | 1B      | 42.94\%             | 42.36\%           | 41.60\%          |
> | 7B      | 53.19\%             | 52.56\%           | 49.33\%          |
>
>
> We have included the above discussion into Appendix H of the revised paper.
>
>  [1] Auer, P. "Finite-time Analysis of the Multiarmed Bandit Problem.", 2002.

---

> > ### Author Response · Authors · 2024-11-23
> >
> > $\textbf{Weakness 2:}$ According to ablation study results, the model's performance is quite sensitive to the sampling threshold of influence. And there are no hyper-parameter selection strategies introduced in the paper. All these make me worry about the robustness of the proposed method to apply in real-world scenarios.
> >
> > $\textbf{Question 1:}$ How do you choose the sampling threshold hyper-parameter in the main experiment? Have you tuned it in a validation set?
> >
> > $\textbf{Response for Weakness 2 and Question 1:}$
> > Thanks for your comments, and we in fact chose the sampling threshold with the help of a validation set.
> > Recap that the sampling threshold of influence ($\tau$) is utilized to determine whether data instances sampled from each cluster should be fed into LLMs for training.
> > If $\tau$ is very small, only data instances with high influence scores are selected.
> > This method of sampling typically results in reduced diversity among instances, which in turn can negatively impact the model performance.
> > If $\tau$ is large enough, many low-quality data instances with low influence scores are selected. This also hurts the model performance, as shown in Figure 4(d) in the paper. Therefore, we agree with the reviewer that choosing a proper $\tau$ is significant.
> >
> > In our experiment, we treated determining the sampling threshold as a hyper-parameter selection problem and solved it with a straightforward but effective method.
> > The basic idea is that we sample some data instances from the candidate dataset. Based on the samples, we try different thresholds, evaluate on the validation test and select the best threshold.
> > We also add an experiment (zooming in Figure 6(a) of the revised paper) to show that the performance of the model is stable around the best selected threshold.
> >
> >
> >
> >
> > To be specific, the hyperparemeter tuning process consists of the following steps.
> > (1) Considering the efficiency issue, we sample about 20\% data instances to form a new candidate dataset from the original one to tune the parameter.
> > (2) We cluster over the candidate dataset, sample some instances from each cluster and compute their influence scores, which are utilized to capture the distribution of influence scores.
> > (3) We rank these scores in descending order and assign them to 20 buckets, so as to derive 20 thresholds, among which we select the best one. For example, the first threshold corresponds to the lowest influence score among the top 5\% instances (the first bucket). The last threshold is the lowest influence score among all instances of buckets.
> > (4) Finally, we select data instances via the $\texttt{Quad}$ algorithm using the above thresholds, evaluate on the validation set, and select the best one.
> >
> > In the table below, the best threshold (i.e., 20\%) is selected using the above method, corresponding to the $4$-th bucket, while other values in the x-axis correspond to the $1$-th, $8$-th, $10$-th bucket respectively, which are far away from the best threshold considering the distribution of actual influence scores.
> > This is the reason why it looks sensitive in Figure 4(d), which highlights the significance of data quality and diversity addressed in our paper.
> >
> > To demonstrate the robustness and effectiveness of the above selection strategy, we add an ablation study by varying the threshold around the best one.  Specifically, as shown in the table below, the x-axis corresponds to the $3$-th, $5$-th, $6$-th buckets, which are neighbors of the best one, and we can observe that the model performance is not sensitive when applying a threshold within these ranges.
> >
> > | Influence Threshold | Bucket | Accuracy |
> > |-|-|-|
> > | 2.60e-3| 3rd    | 42.83%   |
> > | 2.47e-3| 4th    | 42.90%   |
> > | 2.38e-3| 5th    | 42.85%   |
> > | 2.17e-3| 6th    | 42.76%   |
> >
> >
> > We have included the above discussion into Appendix F of the revised paper.

---

> > > ### Comment · Reviewer_Q54u · 2024-11-24
> > > **Thanks for author's response**
> > >
> > > I would like to thank the authors for their response. The additional experiment and more detailed introduction to validation operation improved the soundness of their paper, and I have increased my score.

---

> > > > ### Author Response · Authors · 2024-11-24
> > > > **Thanks for the reviewer's reply**
> > > >
> > > > We appreciate your instant reply for our response! We will improve our paper according to your comments in the final version. Thanks very much!

---

### Official Review · Reviewer_R56B · 2024-11-03

**Soundness:** 3
**Presentation:** 2
**Contribution:** 3
**Rating:** 6
**Confidence:** 3

**Summary:**

This paper presents a new method for selecting LLM pretraining data. The core idea is to measure data quality with diversity. The authors introduce a new approach named Quad, which utilizes attention layers for influence calculation, and clusters data to maintain diversity. Experimental results on the Slimpajama dataset show that Quad surpasses other data selection methods in performance.

**Strengths:**

1. The paper presents a framework that is overall reasonable.
2. The influence calculation with attention layers technique is interesting and could potentially be useful in other settings.
3. Experiments (of the current limited settings) are complete and the authors do release the code for better reproducibility

**Weaknesses:**

The main limitation of this study is its experiment scale is too small (in terms of claiming the method for LLM pretraining). Only 30B out of 627B data are selected for training a small 1.3B decoder-only LLM. In practice, most LLMs will start with at least 7B so how the technique can generalize to reasonably sized LLMs is questionable. Also related to this limitation is the diversity of testing datasets, little (long-text) generation tasks are evaluated.

The second limitation of this study is that the authors do not fully motivate the problem setting. While cleaning noisy data and removing near-duplicate examples are useful for LLM pretraining, I am not fully convinced why you have to select a subset of "quality data" from the full dataset. For example, if we do a coarse-grained filtering of the original 627B to leave around 400-500B data and use all of them to do LLM pretraining, will the model performance better?

Finally, there are a few presentation issues such as the unclear method descriptions (see the below questions) and typos/grammatical issues (just list a few below).

1. Line 042: ”LLMs are generalizable a broad spectrum of downstream tasks”
2. Line 095: “..., which successfully addressing above challenges, achieves”
3. Line 289, “.. will miss massive information of Consequently, …“

**Questions:**

1. Explain figure 1(a) better, what is the x- and y-axis?
2. How do you apply the “JohnsonLindenstrauss Lemma” (line 343)
3. Why only select “30B tokens out of the 627B” (line 360), what if we just do a coarse-grained filtering of the original 627B to leave around 400-500B data (e.g., via near-duplicate removal) and use all of them to do LLM pretraining, will the model performance better?
4. Is the Quad method robust to the clustering algorithms? What features are used in the clulstering?

---

> ### Author Response · Authors · 2024-11-22
>
> $\textbf{Overall Response:}$
>
> Thanks for your insightful comments! Based on your comments, we have added a 7B model, a reference dataset, 7 downstream tasks and 5 additional experiments to demonstrate the scalability and robustness of $\texttt{Quad}$. We additional clarify your concerns about the motivation of our problem setting and some presentation issues in our paper. We promise to include the above discussions into the final version if the paper is accepted.
>
> $\textbf{Limitation 1:}$ The main limitation of this study is its experiment scale is too small (in terms of claiming the method for LLM pretraining). Only 30B out of 627B data are selected for training a small 1.3B decoder-only LLM. In practice, most LLMs will start with at least 7B so how the technique can generalize to reasonably sized LLMs is questionable. Also related to this limitation is the diversity of testing datasets, little (long-text) generation tasks are evaluated.
>
> $\textbf{Response for the first limitation:}$
> We thank your comments for improving our paper. Based on your valuable suggestions, we have added experiments by enlarging the model size and the diversity of downstream tasks, so as to further validate the effectiveness of $\texttt{Quad}$.
>
> | Methods  | Random | Qurating | MATES  | Quad   |
> |-|-|-|-|-|
> | Accuracy | 49.97 | 52.18| 51.96 | 53.19 |
> | FLOPs    | 263.1  | 375.6    | 313.3  | 307.6  |
>
>
>
> To be specific, for the model size, we enlarge the model size from 1.3B to 7B and select 100B tokens from Slimpajama dataset.
> As shown in the table above, we can observe that $\texttt{Quad}$ still performs better than other baselines on accuracy because we select high quality data considering the diversity.
> What's more, $\texttt{Quad}$ also has a good scalability compared with state-of-the-art baselines (Qurating and MATES) because we use the K-FAC technique to accelerate the influence computation and MAB framework to quickly identify the data instances that are beneficial to the downstream tasks.
>
> For testing datasets, we add two downstream tasks (i.e., WikiText [1] and HelloBench [2]) about long-text generation tasks. As shown in the table below, on the candidate dataset Slimpajama, for the 7B model, we can observe that $\texttt{Quad}$ still outperforms other baselines because we simultaneously consider the quality and diversity in data selection.
> We also add five downstream tasks above the math in the next response to further demonstrate that we can well generalize to various tasks.
>
> |           | Random | Qurating | MATES | Quad  |
> |-|-|-|-|-|
> | WikiText  | 16.73  | 15.36    | 15.51 | 14.81 |
> | HelloBench| 3.81   | 5.37     | 4.39  | 6.33  |
>
> Moreover, we also add the Openwebmath dataset as another new reference dataset to train a 7B model on the Slimpajama dataset, which aims at selecting data to improve the mathematical skill of the model. Correspondingly, we also include GSM8K[3], MATH[4], OCW[5], SAT[6], and MMLU STEM[7] as new downstream tasks to evaluate the model's mathematical capabilities, with the final accuracy calculated as the average score across these tasks. As shown in the table below, $\texttt{Quad}$ still demonstrates superior model performance compared to all baselines.
>
> | Methods   | Random | DSIR  | Semdedup | PPL   | Qurating | MATES  | Quad   |
> |-|-|-|-|-|-|-|-|
> | Accuracy  | 33.20 | 33.05| 33.70   | 34.79| 35.31   | 35.20 | 36.71 |
>
>
> We have included the above discussion into Appendix G and I of the revised paper.
>
>  [1] Merity, Stephen, et al. "Pointer sentinel mixture models." arXiv preprint arXiv:1609.07843 (2016).
>
>  [2] Que, Haoran, et al. "Hellobench: Evaluating long text generation capabilities of large language models." arXiv preprint arXiv:2409.16191 (2024).
>
>  [3] Cobbe, Karl, et al. "Training verifiers to solve math word problems." arXiv preprint arXiv:2110.14168 (2021).
>
>  [4] Hendrycks, Dan, et al. "Measuring mathematical problem solving with the math dataset." arXiv preprint arXiv:2103.03874 (2021).
>
>  [5] Lewkowycz, Aitor, et al. "Solving quantitative reasoning problems with language models." NeurIPS, 2022.
>
>  [6] Azerbayev, Zhangir, et al. "Llemma: An open language model for mathematics." arXiv preprint arXiv:2310.10631 (2023).
>
>  [7] Hendrycks, Dan, et al. "Measuring massive multitask language understanding." arXiv preprint arXiv:2009.03300 (2020).

---

> ### Author Response · Authors · 2024-11-22
>
> $\textbf{Limitation 2:}$ The second limitation of this study is that the authors do not fully motivate the problem setting. While cleaning noisy data and removing near-duplicate examples are useful for LLM pretraining, I am not fully convinced why you have to select a subset of ``quality data'' from the full dataset. For example, if we do a coarse-grained filtering of the original 627B to leave around 400-500B data and use all of them to do LLM pretraining, will the model performance better?
>
> $\textbf{Question 3:}$ Why only select ``30B tokens out of the 627B'' (line 360), what if we just do a coarse-grained filtering of the original 627B to leave around 400-500B data (e.g., via near-duplicate removal) and use all of them to do LLM pretraining, will the model performance better?}
>
>
>
> $\textbf{Response for the second Limitation and Question 3:}$
> In this response, we first discuss the advantage of using  $\texttt{Quad}$ to select data instances. We then run experiments to validate the argument.
>
> Overall, we agree with the reviewer that data cleaning  is likely to enhance the model performance. However, the obtained data after cleaning is still large-scale, which is prohibitively expensive to train, but not always necessary. This is because a subset of judiciously selected data can achieve competitive or even superior model performance compared with training with all clean data. The reasons are two-fold. First, $\texttt{Quad}$ is a model-aware data selection method, which uses the data influence to measure the impact of each data instance on current pretraining progress,  but coarse-grained data cleaning methods fail to capture the model's current state. Second, $\texttt{Quad}$ also considers the diversity for data selection, and thus the selected small part of data instances can well cover different characteristics of downstream tasks.
>
> Overall, model-aware data selection is of great significance because (1) it is customized to the current pretraining progress, which can greatly improve the model performance with a small number of data instances; and (2) the model can be trained over these selected small amounts of data, thus greatly improving the training efficiency.
>
> Next, we run two new experiments to validate this.
>
> First, we would like to clarify that the SlimPajama dataset used in our experiment has already  undergone multiple rounds of coarse-grained data cleaning processes [1], including  short documents filtering, deduplication, minhash generation, etc.
> Therefore, to demonstrate the significance of the data selection method, we compare the performance of $\texttt{Quad}$ and directly training with clean data with different data sizes with 1.3B and 7B models, evaluating on the long-text generation tasks  mentioned in the last response.  As shown in the table below, we can observe that `Quad`-1.3B-100B outperforms `Clean`-1.3B-100B because `Quad` judiciously selects data that benefits the current pretraining progress while the latter one just selects 100B data randomly from the clean  SlimPajama dataset without considering the current pretraining progress.  Because of the similar reason, `Quad`-1.3B-100B achieves comparable model performance with `Clean`-1.3B-400B, which saves 75\% of computation costs.  We can also observe that training on 7B model has a similar result.
>
>
> | Methods| Accuracy | FLOPs  |
> |-|-|-|
> | `Clean`-1.3B-100B| 44.41| 58.1|
> | `Clean`-1.3B-400B| 47.66| 232.5|
> | `Quad`-1.3B-100B| 47.88| 72.3|
> | `Clean`-7B-100B| 49.73|263.1|
> | `Clean`-7B-400B| 53.11|1063.7|
> | `Quad`-7B-100B| 53.19| 307.6 |
>
>
>
> Moreover, we add another dataset Openwebmath [2] as our new reference dataset. It is based on the Slimpajama dataset, which aims at selecting data to improve the mathematical skill of the model, i.e., a specific domain. We use GSM8K, MATH, OCW, SAT, MMLU STEM as the downstream tasks and the final accuracy is computed the average score of these downstream tasks.
> As shown in the table below, we can observe that random selection of 400B data instances does not perform better than selecting 100B by $\texttt{Quad}$ on downstream mathematical evaluation datasets.
> This is because the 400B data instances contain a significant amount of information irrelevant to mathematics, which degrades the model performance, while $\texttt{Quad}$ is a model-aware method that accurately identifies the data that benefits the current pretraining progress.
>
> |Methods| Accuracy|
> |-|-|
> | `Clean`-7B-100B| 33.20|
> | `Clean`-7B-400B| 35.01|
> | `Quad`-7B-100B| 36.76|
>
> We have included the above discussion into Appendix G of the revised paper.
>
>  [1] https://cerebras.ai/blog/slimpajama-a-627b-token-cleaned-and-deduplicated-version-of-redpajama
>
>  [2] Paster, Keiran, et al. "Openwebmath: An open dataset of high-quality mathematical web text." arXiv preprint arXiv:2310.06786 (2023).

---

> > ### Author Response · Authors · 2024-11-23
> >
> > $\textbf{Weakness 3:}$ Finally, there are a few presentation issues such as the unclear method descriptions (see the below questions) and typos/grammatical issues.
> >
> > $\textbf{Response:} $ Thanks! We have fixed these problems and carefully revised our paper.
> >
> > $\textbf{Question 1:}$ Explain figure 1(a) better, what is the x- and y-axis?
> >
> > $\textbf{Response:}$ We utilize the t-SNE [1] technique to transform the high-dimensional data collected from the candidate dataset into two-dimensional data, and then we  plot  Figure 1(a) in our paper. In this depiction, the two-dimensional plane represents the x- and y-axis in the figure.
> >
> > $\textbf{Question 2:}$ How do you apply the "JohnsonLindenstrauss Lemma'' (line 343)
> >
> > $\textbf{Response:}$
> > Thanks for your comments! The Johnson-Lindenstrauss Lemma[2] is utilized to map data instances in a high-dimensional space to a low-dimensional space through a linear transformation, while approximately preserving the Euclidean distances between the data instances.
> > In Equation (10), $ H_{att}^{-1} $ denotes the inverse Hessian matrix on the attention layer and $v_{att}$ denotes the gradient vector of reference data instances on the attention layer.
> > After computing $H_{att}^{-1}v_{att}$ using the K-FAC method, we leverage the Lindenstrauss Lemma to reduce its dimensionality, thereby  conserving memory.
> > We achieve this by projecting $H_{att}^{-1}v_{att}$ onto a $d$-dimensional space, resulting in $Q^T H_{att}^{-1}v_{att}$.
> > In the Lemma, each element of $Q$ is drawn from $\mathbb{R}^{P \times d}$, with $P$ representing the original dimensionality and $d$ denoting the reduced dimensionality. To calculate the influence function (i.e., $I_{\theta_{att}} = -\nabla L(\theta_{att}, z)H_{att}^{-1}v_{att}$.), we also have to  apply the same dimensionality reduction method to transform $\nabla L(\theta_{att},z)$ into $Q^T \nabla L(\theta_{att},z)$, so as to avoid avoid excessive memory usage.
> >
> > $\textbf{Question 4:}$ Is the $\texttt{Quad}$ method robust to the clustering algorithms? What features are used in the clustering?
> >
> > $\textbf{Response for Question 4:} $
> > Thanks for your valuable feedback. In response to your question, we have added an ablation study regarding the clustering algorithms, which has been included in our revised paper. We evaluate the performance of different typical clustering methods, including GMM [3], DBSCAN [4] and MeanShift [5] to cluster data points in the candidate dataset. Considering that the clustering result is affected by the parameters of the clustering algorithms, we can use different methods to select the most appropriate parameters. For GMM, we can use the AIC score[6] to determine the appropriate number of components. For DBSCAN, there are 2 key parameters: (1) $eps$(the radius of a neighborhood w.r.t. some data points) (2) $minPts$ (a data instance is considered as a core point if at least $minPts$ data points are within $eps$ of it). They can be set using the method in [7]. Mean-Shift is a centroid-based method that updates the centroids to be the mean of the instances within a given region. The size of the region is controlled by $bandwidth$, which can be set by the bandwidth estimation [8]. In this experiment, the experimental settings of the 1.3B model align with the configurations noted in our original paper. The results in the above table  indicate that our $\texttt{Quad}$ method demonstrates robustness regardless of the clustering algorithms used.
> >
> > |Clustering Methods|K-MEANS| DBSCAN | GMM| MeanShift |
> > |-|-|-|-|-|
> > |Accuracy|42.94|42.89|42.93| 42.90|
> >
> > For the features, we utilize the embedding of each data instance as the feature. Specifically, we leverage a frequently used, BERT-based model $\texttt{BAAI/bge-large-en-v1.5}$[9] to calculate the embedding of each data instance and then clustering them (see line 343 of the original paper).
> >
> > We have included the above discussion into Appendix D of the revised paper.

---

> > > ### Author Response · Authors · 2024-11-23
> > >
> > > [1] Van der Maaten, Laurens, and Geoffrey Hinton. ``Visualizing data using t-SNE.'' Journal of machine learning research, 2008.
> > >
> > >  [2] Johnson, W. B. and Lindenstrauss, J. Extensions of lipschitz mappings into hilbert space. Contemporary mathematics, 26:189–206, 1984.
> > >
> > >  [3] Figueiredo, Mario A. T., and Anil K. Jain. "Unsupervised learning of finite mixture models." IEEE Transactions on pattern analysis and machine intelligence, 2002.
> > >
> > >  [4] Ester, Martin, et al. "A density-based algorithm for discovering clusters in large spatial databases with noise." KDD, 1996.
> > >
> > >  [5] Cheng, Yizong. "Mean shift, mode seeking, and clustering." IEEE transactions on pattern analysis and machine intelligence, 1995.
> > >
> > >  [6]  Aho, Ken, DeWayne Derryberry, and Teri Peterson. "Model selection for ecologists: the worldviews of AIC and BIC." Ecology 2014
> > >
> > >  [7] Schubert, Erich, et al. "DBSCAN revisited, revisited: why and how you should (still) use DBSCAN." ACM Transactions on Database Systems 2017
> > >
> > >  [8] Sklearn. 2022. https://scikit-learn.org/stable/modules/generated/sklearn.cluster.estimate\_bandwidth.html
> > >
> > >  [9] https://huggingface.co/BAAI/bge-large-en-v1.5

---

> > > > ### Author Response · Authors · 2024-11-25
> > > > **Looking forward to your reply.**
> > > >
> > > > Dear reviewer R56B,
> > > >
> > > > Greetings from the authors!
> > > >
> > > > We would like to express our sincere gratitude for your insightful comments.
> > > >
> > > > Regarding the experiment scale, problem setting and other concerns, we have conducted multiple experiments and included these experiments and analysis in the revised paper. We will include them in the final version and provide the link to our GitHub repository upon acceptance.
> > > >
> > > > If you have any question for our paper, please feel free to point out and we will try to address it quickly. Thanks for your time and looking forward to your reply!

---

> ### Comment · Reviewer_R56B · 2024-11-25
>
> Thanks for the responses. They do improve the paper and thus I increase the score by 2 for the additional experiments and discussions. However, I still think my core questions are not addressed.
>
> For the topic of "pre-training", the experiment scale in this paper (even considering the 7B experiments in the rebuttal) is not sufficient. How well the Quad method generalize to larger model is questionable. When the model becomes large enough, do we still need to select a subset of good examples instead of focusing on getting more high quality data? I understand due to practical reasons (e.g., compute budgets), it's hard to do large scale experiments in some settings, but again, for the "pre-training" research, this is a challenge we have to face.
>
> Second, the authors claim one advantage of Quad is "model-aware data selection" that is customized to the downstream tasks. However, I think the goal of "pre-training" is to get a good foundation (unaligned) model that is good for later-stage post-training/alignment. If we just want to get good scores for a few downstream tasks, we don't do it in the pre-training stage.
>
> Third, related to the problem setting, in real-world, when will people do 7B-scale model pre-training from scratch? Usually people either start from a pre-trained checkpoint then do post-training/alignment (for a few specific tasks), or do continuous pre-training/tailpatch training (for knowledge enhancement). If we really need a small scale pre-train ckpt, most likely we will distill a large pre-train ckpt to a smaller model. So how useful this problem setting should be further discussed.
>
> Finally, it would be interesting to show Quad-7B-400B and compare with Clean-7B-400B and see the performance gap.
>
> To summarize, I appreciate authors' due diligence on improving the paper quality, but I still think there are many questions need to be answered in this work.

---

> > ### Author Response · Authors · 2024-11-30
> >
> > $\textbf{Overview:}$
> >
> > Wishing you a happy and blessed Thanksgiving! We appreciate the reviewer for improving the rating and offering additional insightful comments. Our replies addressing the comments are presented below from the following perspectives.
> >
> > $\textbf{Weakness 1}$
> > For the topic of "pre-training", the experiment scale in this paper (even considering the 7B experiments in the rebuttal) is not sufficient. How well the Quad method generalize to larger model is questionable. When the model becomes large enough, do we still need to select a subset of good examples instead of focusing on getting more high quality data? I understand due to practical reasons (e.g., compute budgets), it's hard to do large scale experiments in some settings, but again, for the "pre-training" research, this is a challenge we have to face.
> >
> > $\textbf{Response:}$
> >
> > Thanks for your comments. First, we agree with the reviewer that the scalability of a pre-training data selection method is a challenging and significant aspect. Although the trend from 1.3B to 7B  (the largest experiment scale compared with state-of-the-art pre-training data selection baselines [1][2][3] that just conduct experiments on 1.3B) indicates the robustness of our observations,  it remains unclear how well our methods scale up to production level models with billions of parameters and trillions of pre-training tokens, which is indeed a challenge of most exploratory LLMs research works having to face. For example, many works ($e.g.,$ [6][7] for data mixing, [8][10] for LLM pruning, [9][11] for optimization algorithms) focusing on different aspects of LLMs pre-training conduct experiments on model size around 7B, which is likely to provide insights that can benefit larger LLMs training. Unfortunately, due to the limited computing resources, we cannot conduct more scalability experiments within the rebuttal phase, but we believe that our method can offer insights for the community on this field, and we have included the limitation in the revised paper, leaving a promising future work.
> >
> >
> > Second, we would like to clarify that training large LLMs still prefers our fine-grained data selection strategy, comparing to simple coarse-grained low-quality data filtering.
> > In the technical report of GPT-3 [4], the training data was 570GB selected from a 45TB data pool ($i.e.$ CommonCrawl). It reported that lightly filtering the data is not sufficient, and thus they built models to measure the quality of each data instance based on high-quality reference datasets.  In addition, as illustrated in the technical report of $\texttt{Llama-3}$ [5], it pointed out that besides coarse-grained data filtering methods like text extraction, cleaning and de-duplication, fine-grained quality filtering is still necessary. Similar to $\texttt{GPT-3}$ [4], they built different models for data selection, scoring each data instance based on reference datasets. What's more, in academia, `Qurating` (ICML 2024) and `MATES` (NeurIPS 2024) respectively use the high-performance LLM and influence function to select a subset of data from a candidate dataset. These methods are also considered as different types of data selection models. Our `Quad` uses influence scores to select data taking both the quality and diversity into account, achieving superior results than the above state-of-the-art baselines.
> >
> >  [1] Yu, Zichun, Spandan Das, and Chenyan Xiong. "MATES: Model-Aware Data Selection for Efficient pre-training with Data Influence Models." NeurIPS, 2024.
> >
> >  [2] Wettig, Alexander, et al. "Qurating: Selecting high-quality data for training language models." arXiv preprint arXiv:2402.09739 (2024).
> >
> >  [3] Xie, Sang Michael, et al. "Data selection for language models via importance resampling." NeurIPS, 2023.
> >
> >  [4] Brown, Tom B. "Language models are few-shot learners." arXiv preprint arXiv:2005.14165 (2020).
> >
> >  [5] Dubey, Abhimanyu, et al. "The llama 3 herd of models." arXiv preprint arXiv:2407.21783 (2024).
> >
> >  [6] Xie, Sang Michael, et al. "Doremi: Optimizing data mixtures speeds up language model pre-training." NeurIPS, 2024.
> >
> >  [7] Albalak, Alon, et al. "Efficient online data mixing for language model pre-training." R0-FoMo: Robustness of Few-shot and Zero-shot Learning in Large Foundation Models, 2023.
> >
> >  [8] Xia, Mengzhou, et al. "Sheared llama: Accelerating language model pre-training via structured pruning." NeurIPS, 2023.
> >
> >  [9] Chen, Xiangning, et al. "Symbolic discovery of optimization algorithms." NeurIPS, 2024.
> >
> >  [10] Ma, Xinyin, Gongfan Fang, and Xinchao Wang. "Llm-pruner: On the structural pruning of large language models." NeurIPS, 2023.
> >
> >  [11] Liu, Hong, et al. "Sophia: A scalable stochastic second-order optimizer for language model pre-training."ICLR, 2024.

---

> > > ### Author Response · Authors · 2024-11-30
> > >
> > > $\textbf{Weakness 2}$
> > > Second, the authors claim one advantage of Quad is "model-aware data selection" that is customized to the downstream tasks. However, I think the goal of "pre-training" is to get a good foundation (unaligned) model that is good for later-stage post-training/alignment. If we just want to get good scores for a few downstream tasks, we don't do it in the pre-training stage.
> > >
> > > $\textbf{Response:}$
> > >
> > > Sorry for the confusion. Actually, in the rebuttal,  what we wanted to express is that `Quad` supports data selection for both pre-training from scratch and continuous pre-training in specific domains. However, due to time constraints, miscommunication occurred, leading to inaccurate statement ("customized to downstream tasks''). Note that in the submitted paper manuscript, we in fact did not mention that our method is customized for downstream tasks. Now, we have modified the previous rebuttal and corrected the presentation issue, as shown in Section 3.1 of our revised paper.
> > >
> > > Specifically, unlike heuristic methods that select data only considering the characteristics of the data itself, model-aware data selection considers the impact of each data instance on the model, $i.e.,$ selecting the most effective data for the current (continuous) pre-training.
> > >
> > > For pre-training, we use LAMBADA [1] as our reference set, which is commonly used as a validation benchmark[13][14][15][16] to evaluate the ability of LLMs for language comprehension and language generation within a broad discourse context, not only just for a few downstream tasks. Moreover, to ensure that the pre-trained LLM is unaligned, in the experiment, we selected a variety of downstream tasks (e.g., CommonSenseQA[2], BoolQ[3], ARC-Challenge[4], and others) that are commonly used to evaluate  LLMs such as Llama-3[5] and GPT-4[6]. These tasks cover a broad range of essential capabilities required for pre-trained LLMs.
> > >
> > > For continuous pre-training, we conduct an additional experiment that uses Llama2-7B as the base model, based on which we use the SlimPajama as the candidate dataset, and select data from it to enhance mathematical capabilities.
> > > For evaluation, we employ GSM8K [8], MATH [9], OCW [10], SAT [11], and MMLU STEM [12] as downstream tasks, with the final accuracy calculated as the average score across these tasks. We use Openwebmath[7] for the  reference set.
> > > As shown in the table below, we can observe that random selection of 400B data instances does not perform better than selecting 100B by `Quad` on downstream math datasets. This is because the 400B data instances contain a significant amount of information irrelevant to mathematics, which degrades the model performance.
> > > However, `Quad` is a model-aware method that can identify the data instances beneficial to the current pre-training process.
> > >
> > >
> > > | Methods             | Accuracy  |
> > > |-|-----------|
> > > | `Clean`-7B-100B     | 33.20%    |
> > > | `Clean`-7B-400B     | 35.01%    |
> > > | `Quad`-7B-100B      | 36.76%    |
> > >
> > > [1] The LAMBADA dataset: Word prediction requiring a broad discourse context
> > >
> > > [2] Talmor, Alon, et al. "Commonsenseqa: A question answering challenge targeting commonsense knowledge." arXiv preprint arXiv:1811.00937 (2018).
> > >
> > > [3] Clark, Christopher, et al. "BoolQ: Exploring the surprising difficulty of natural yes/no questions." arXiv preprint arXiv:1905.10044 (2019).
> > >
> > > [4] Clark, Peter, et al. "Think you have solved question answering? try arc, the ai2 reasoning challenge." arXiv preprint arXiv:1803.05457 (2018).
> > >
> > > [5] Dubey, Abhimanyu, et al. "The llama 3 herd of models." arXiv preprint arXiv:2407.21783 (2024).
> > >
> > > [6]  Language models are few-shot learners.
> > >
> > > [7] Paster, Keiran, et al. "Openwebmath: An open dataset of high-quality mathematical web text." arXiv preprint arXiv:2310.06786 (2023).
> > >
> > > [8] Cobbe, Karl, et al. "Training verifiers to solve math word problems." arXiv preprint arXiv:2110.14168 (2021).
> > >
> > > [9] Hendrycks, Dan, et al. "Measuring mathematical problem solving with the math dataset." arXiv preprint arXiv:2103.03874 (2021).
> > >
> > > [10] Lewkowycz, Aitor, et al. "Solving quantitative reasoning problems with language models." NeurIPS, 2022.
> > >
> > > [11] Azerbayev, Zhangir, et al. "Llemma: An open language model for mathematics." arXiv preprint arXiv:2310.10631 (2023).
> > >
> > > [12] Hendrycks, Dan, et al. "Measuring massive multitask language understanding." arXiv preprint arXiv:2009.03300 (2020).
> > >
> > > [13] Yu, Zichun, Spandan Das, and Chenyan Xiong. "MATES: Model-Aware Data Selection for Efficient pre-training with Data Influence Models." NeurIPS, 2024.
> > >
> > > [14] Brown, Tom B. "Language models are few-shot learners." arXiv preprint arXiv:2005.14165 (2020).
> > >
> > > [15] Chowdhery, Aakanksha, et al. "Palm: Scaling language modeling with pathways." Journal of Machine Learning Research 24.240 (2023): 1-113.
> > >
> > > [16] Hoffmann, Jordan, et al. "An empirical analysis of compute-optimal large language model training." NeurIPS, 2022.

---

> > > > ### Author Response · Authors · 2024-11-30
> > > >
> > > > $\textbf{Weakness 3}$
> > > >
> > > > Third, related to the problem setting, in real-world, when will people do 7B-scale model pre-training from scratch? Usually people either start from a pre-trained checkpoint then do post-training/alignment (for a few specific tasks), or do continuous pre-training/tailpatch training (for knowledge enhancement). If we really need a small scale pre-train ckpt, most likely we will distill a large pre-train ckpt to a smaller model. So how useful this problem setting should be further discussed.
> > > >
> > > > $\textbf{Response:}$
> > > >
> > > > Thanks for the valuable feedback from the reviewer.
> > > > We would like to clarify that for large Internet enterprises,  it is common to train their own LLMs from scratch. For small companies and researchers, they often conduct continuous pre-training or supervised fine-tuning (SFT) based on pre-trained checkpoints, while for end users, they typically interact with LLMs through prompts.
> > > > In real-world, although there already exist well-performing LLMs ($e.g.,$ Llama-7B), many big IT companies still choose to train their own large models from scratch, such as Mistral's $\texttt{Mistral-7B}$ [1] and Google's $\texttt{Gemini-7B}$ [2].
> > > > Even when training $\texttt{Llama-3}$ [3], Meta did not continuously pre-train $\texttt{Llama-2}$ [4] but instead retrained from scratch.
> > > > However, we agree with the reviewer that to meet the domain specific needs, people always conduct continuous pre-training to enhance the generic LLMs. This also readily benefits from our data selection strategy `Quad`.
> > > >
> > > > [1] Jiang A Q, Sablayrolles A, Mensch A, et al. Mistral 7B[J]. arXiv preprint arXiv:2310.06825, 2023.
> > > >
> > > > [2] Team, Gemini, et al. "Gemini: a family of highly capable multimodal models." arXiv preprint arXiv:2312.11805 (2023).
> > > >
> > > > [3] Dubey, Abhimanyu, et al. "The llama 3 herd of models." arXiv preprint arXiv:2407.21783 (2024).
> > > >
> > > > [4] Touvron, Hugo, et al. "Llama 2: Open foundation and fine-tuned chat models." arXiv preprint arXiv:2307.09288 (2023).
> > > >
> > > >
> > > > $\textbf{Weakness 4}$
> > > >
> > > > Finally, it would be interesting to show `Quad`-7B-400B and compare with `Clean`-7B-400B and see the performance gap.
> > > >
> > > > $\textbf{Response:}$
> > > >
> > > > Thanks for your valuable suggestion to improve our paper. As the reviewer suggested, we have added an experiment of `Clean`-7B-400B. Moreover, we have run experiments to compare `Clean` with `Quad` on the checkpoints of pre-training Llama2-7B on SlimPajama all the way to 600B . The results are shown in the below table.  We can observe that `Quad`-7B-400B outperforms `Clean`-7B-400B by 2.21 \%, confirming that simple heuristics are less effective in discovering high-quality (low-quality) data. However, as the training data size increases, the average quality of the data selected by `Clean` becomes increasingly similar to that of `Quad`, leading to a smaller accuracy gap between the `Quad` and `Clean` methods. Note that to select a larger amount of data (e.g., 600B tokens), we lower the influence threshold in  `Quad`. Otherwise, in practice, there is no need for `Quad` to select such a large amount of data.
> > > >
> > > > From these checkpoints, we can observe that by using the `Quad` method, a high accuracy can be quickly achieved with the first 300B data, and then the accuracy remains stable. This shows that using `Quad` to select 300B data is able to produce a model with an accuracy comparable to the model trained on 600B data.  Therefore, leveraging `Quad`, we can start with a small data size for pre-training and increase the size until the accuracy remains stable. This largely improves the data efficiency, while not sacrifices the accuracy, thus saving a substantial amount of computational costs.
> > > >
> > > > |Tokens(B)|100|200|300|400|500|600|
> > > > |-|-|-|-|-|-|-|
> > > > |`Clean`|49.73|51.19|52.36|53.11|54.21|55.15|
> > > > |`Quad`|53.19|54.67|55.19|55.31|55.17|55.23|
> > > >
> > > > In addition, we have conducted a similar experiment under the continuous pre-training setting. It uses Llama2-7B as the base model, SlimPajama as the candidate dataset and Openwebmath as the reference set. The table below shows the results $w.r.t.$ the training checkpoints ranging from 100B to 600B tokens. We can observe that when data is selected randomly from SlimPajama, the performance gains of the LLM on the downstream evaluation set remain relatively small, even when using the 600B tokens. The reason is that random selection selects many instances irrelevant to math. For `Quad`, we can observe that selecting 200B data achieves the best accuracy, much higher than any checkpoint of `Clean`. With the increasing number of tokens, the performance of `Quad` decreases because data relevant to math has already been selected in the candidate dataset in early checkpoints.
> > > > Thus, for continuous pre-training, `Quad` can improve both the effectiveness and efficiency in a way same to starting from a small subset and increasing the subset size.
> > > >
> > > > |Tokens(B)|100|200| 300|400|500|600|
> > > > |-|-|-|-|-|-|-|
> > > > |`Clean`|33.20|34.31|34.87|35.01|34.91|35.30|
> > > > |`Quad`|36.76|39.13|38.51|37.03|35.87|35.37|

---

> ### Author Response · Authors · 2024-12-02
> **Looking forward to your reply**
>
> Dear reviewer $\texttt{R56B}$
>
> Thank you for your thoughtful efforts in reviewing our work! As the discussion period nears its end, we kindly hope you can take a moment to review our rebuttal. Looking forward to your feedback!
>
> Best Wishes!
>
> ICLR Authors

---

> ### Comment · Reviewer_R56B · 2024-12-02
>
> Thanks for the detailed response and I appreciate authors' efforts on addressing some of my concerns. I go over more related literatures in the "pre-training data selection" field and agree that 7B scale experiments are a generally acceptable scale in academia. Therefore, I will increase the score by 1 and am OK to see this paper accepted. Hopefully it can benefit the community and inspire for more data-centric work in the field.
>
> Below are some of my add-on thoughts and suggestions on this work.
>
> First, I think the authors should further emphasize the continuous pre-training part and conduct more experiments. The current experiments on math benchmark is not really sufficient. It only tests the model's math capabilities on 5 datasets. In some sense, this is already very close to the post-training stage and we are aligning model for more math use cases. The ideal experiment setting is to check model performance on tasks that require new knowledge from continuous pretrain dataset, meanwhile reporting the model performance using pretrain metrics like perplexity or few-shot ICL numbers. This leads to another question, when your reference dataset is huge, will the influence score computation very small (or even possible if large dataset is coupled with large model).
>
> Second, the practical meaning of problem setting -- "when will people do 7B-scale model pre-training from scratch?" is still not really answered. For big tech companies like Google, Meta, Nvidia, they will not really train such a small LLM from scratch. Instead, they will have computes to train an O(100B) large model and distill it into smaller models. For example, Gemma-2 2B and 9B models are distilled from 27B model (https://arxiv.org/html/2408.00118v1), llama 3.2 1B model is distilled from llama 3.1 8B model, and Nvidia distills llama-3.1 8B to llama-3.1-Minitron 4B (https://developer.nvidia.com/blog/how-to-prune-and-distill-llama-3-1-8b-to-an-nvidia-llama-3-1-minitron-4b-model). I personally think more discussions on continuous pre-training is more meaningful and valuable.
>
> Finally, I would suggest authors expanding its K-FAC discussion with more discussions in the appendix, maybe adding some pseudo code.
>
> To summarize, even this work has some limitations and weaknesses, I agree it follows the subfield norms and standards and thus can be accepted.

---

> > ### Author Response · Authors · 2024-12-03
> >
> > We appreciate your reply for our response!
> >
> > We will follow your suggestions to consider comprehensive experiments ($w.r.t.$ continuous pretraining) in our work. Also, we will include more detailed discussion related to K-FAC in the appendix.
> >
> > Again, we sincerely thank you for your insightful and detailed comments throughout this process!
> >
> > Best wishes!
> >
> > ICLR Authors

---

### Official Review · Reviewer_yV6J · 2024-11-05

**Soundness:** 3
**Presentation:** 3
**Contribution:** 3
**Rating:** 8
**Confidence:** 4

**Summary:**

The paper presents a method for data selection, namely Quad, which (i) uses the influence function with multi-head attention layers to estimate the potential impact of individual samples on the model's performance; (ii) uses clustering and multi-armed bandits (where each cluster is an arm) to balance between quality and diversity. The paper further proposes acceleration techniques to reduce computational cost and memory usage. The proposed method is applied on the SlimPajama dataset by training a 1.3B model, evaluated on a set of common text benchmarks, and compared with a set of data selection baselines.

**Strengths:**

The paper addresses the important challenge of data selection via a novel method and practical acceleration techniques. Influence function is an expensive operation and the proposed techniques make it more accessible for LLMs where computational overhead is an important factor.

The paper presents a comprehensive evaluation on downstream tasks as well as ablations and statistics to help understand the proposed method.

Despite the method's complexity, the paper is overall clearly written and easy to follow. The authors also attach the code, which helps the community to adapt this method for custom usecases.

**Weaknesses:**

The proposed method requires a complex set of steps and hyperparameters (e.g., $\alpha$,  $\gamma$, and $\tau$ for multi-armed bandit) and, even though there are ablations for some of these, it
might be hard and resource-intensive to develop and tune in practice for diverse applications.

The experimental findings show that all evaluated methods have limited difference in downstream performance and it is not clear whether the observed improvements are statistically significant. It is also concerning that baselines like DSIR show no benefit compared to random selection, which might be contradicting with findings from previous work. Therefore, I would recommend a more thorough discussion behind the empirical benefits of the proposed method and the connection of the findings compared to the previous literature. To better demonstrate the benefits, it might be worth (i) considering more experimental settings, for example with models of varying sizes; (ii) plotting the training and validation losses for each data selection method.

**Questions:**

Could the authors report the results of statistical significance tests?

How does the selection of clustering algorithm and number of cluster affect the performance of the proposed method?

Can the authors clarify how they compute the FLOPS column in Table 1?

---

> ### Author Response · Authors · 2024-11-22
>
> $\textbf{Overall response.}$
>
> Thanks for your insightful feedback! Based on your comments, we have added a 7B model and 6 additional experiments to demonstrate the robustness and effectiveness of $\texttt{Quad}$. We also clarify your concerns about  hyperparameter tuning, the DSIR baseline, significance test, etc. We promise to include the above discussions into the final version if the paper is accepted.
>
> $\textbf{Question 1:}$ Could the authors report the results of statistical significance tests?
>
> $\textbf{Response:}$
>
> Thanks for your suggestions to improve our paper! Based on the reviewer's advice, we run the experiment six times and report the statistics significance tests. The results are as follows:
>
> |Methods|Exp-1|Exp-2|Exp-3|Exp-4|Exp-5|Exp-6| Avg $\pm$ std|
> |-|-|-|-|-|-|-|-|
> |Random|41.52| 41.32 | 41.68| 41.72 | 41.57 | 41.55 | 41.56 $\pm$ 0.13|
> |DSIR|40.59| 40.47 | 40.72  | 40.29 | 40.37 | 40.53 | 40.49 $\pm$ 0.14|
> |PPL|41.35| 41.72| 41.55  | 41.60 | 41.61 | 41.57 | 41.57 $\pm$ 0.11|
> |Semdedup|41.60|41.94| 41.87 | 42.04| 41.91 | 41.81 | 41.86 $\pm$ 0.14|
> |Qurating|42.03| 41.89| 41.81 | 42.21| 42.13| 42.01 | 42.01 $\pm$ 0.13|
> |MATES|42.10| 41.76| 42.20  | 41.70| 41.81| 41.93  | 41.92 $\pm$ 0.18|
> |Quad (ours)|43.11 |42.91 | 43.03| 43.02| 42.87| 42.94 | 42.98 $\pm$ 0.08|
>
> Based on the above results, we perform a significance test utilizing t-tests.  We make the following two hypothesis. (1) Null hypothesis ($H_0$): The average accuracy of $\texttt{Quad}$ does not exceed the average of baselines; and (2) Alternative hypothesis ($H_1$): The average performance of $\texttt{Quad}$ exceeds that of the baselines.
> We set the confidence level as 99\%, with a significance level of $\alpha=0.01$.
> The p-value represents the probability of observing the test statistic under the assumption that  $H_0$ is true. A small p-value indicates a lower likelihood of observing the current result if  $H_0$ holds.
>
> | t_test | Random| DSIR | PPL | Semdedup | Qurating | MATES |
> |-|-|-|-|-|-|-|
> | p-value |7e-10|5e-12| 2e-10| 1e-08 | 4e-08   | 2e-07 |
>
> For each baseline, we can observe $p<\alpha$, and thus we reject $H_0$ and accept $H_1$, indicating that the average performance of $\texttt{Quad}$ is greater than that of  baselines. Since the confidence level of t-test surpasses 99\%, we can conclude that the superiority of $\texttt{Quad}$ compared to other baselines is statistically significant. We have included the above discussion in Appendix C of the revised paper.
>
> $\textbf{Question 2:}$ How does the selection of clustering algorithm and number of cluster affect the performance of the proposed method?
>
> $\textbf{Response:}$
>
> Thanks for your suggestion. In response to your question, we add an ablation study regarding the clustering algorithms and number of clusters.
>
> Firstly, we add an ablation study by varying the number of clusters, as shown in the table below.  Using the Slimpajama dataset, the Elbow [1] method selected $k=10,000$ as the optimal number of clusters in our original paper. From the following table we can observe that the model consistently performs well when the cluster number is close to 10,000. Conversely, a very small number of clusters (i.e., $k=1,000$) leads to poor performance due to the high variance of influence scores in each cluster. Thus, the sampled instances cannot well represent the cluster. Similarly, when the cluster number is too high(i.e., $k=500,000$), there will be many clusters that are in fact contain similar data instances, and thus it is relatively hard to explore diverse clusters, thereby leading to the performance degradation.
>
> |Cluster Numbers|1000|5000|10000|20000|50000|100000|
> |-|-|-|-|-|-|-|
> |Accuracy|42.73|42.90|42.94|42.93|42.87|42.77|
>
> Moreover, we evaluate the performance of several typical clustering methods including GMM[2], DBSCAN[3] and MeanShift[4].
> Considering that the clustering results are affected by the parameters of  clustering algorithms, we use different methods to select proper parameters. For GMM, we can use the AIC score [5] to determine the appropriate number of components. For DBSCAN, there are 2 key parameters: (1) $eps$(the radius of a neighborhood w.r.t. some data points) and (2) $minPts$ (a data point is considered as a core point if at least $minPts$ data points are within $eps$ of it). They can be set using the method in [6]. Mean-Shift is a centroid-based method that updates the centroids to be the mean of the points within a given region. The size of the region is controlled by $bandwidth$, which can be set by the estimation of the bandwidth [7].
> In this set of experiments, the experimental settings of pretraining 1.3B model are consistent with those reported in our paper. The above result shows that $\texttt{Quad}$  is not sensitive to the clustering algorithms.
>
> |Clustering Methods|K-MEANS|DBSCAN|GMM |MeanShift|
> |-|-|-|-|-|
> |Accuracy|42.94|42.89|42.93|42.90|
>
> We have included the above discussion into Appendix D of the revised paper.

---

> > ### Author Response · Authors · 2024-11-22
> >
> > $\textbf{Question 3:}$ Can the authors clarify how they compute the FLOPS column in Table 1?
> >
> > $\textbf{Response:}$
> >
> > FLOPs is the number of floating point operations performed by GPUs. Many state-of-the-art methods [8,9,10] use it to measure the consumption of  GPU computing resources. In our experiments, FLOPs is collected directly in the data selection process using the Python code:
> >
> > ```python
> > import torch
> > import torch.nn as nn
> > from torch.profiler import profile, ProfilerActivity
> >
> > model = nn.Linear(1024, 512).cuda()
> > input_data = torch.randn(128, 1024).cuda()
> >
> > with profile(activities=[ProfilerActivity.CPU, ProfilerActivity.CUDA], with_flops=True) as prof:
> >     model(input_data)
> >
> > print(prof.key_averages().table(sort_by="flops", row_limit=10))
> > ```
> > $\textbf{Weakness 1}$
> >
> > $\textbf{Response:}$
> > We appreciate your feedback! We agree with the reviewer regarding the need to fine-tune some hyperparameters in $\texttt{Quad}$. However, this task is manageable. Typically, we can sample a subset of the candidate set, based on which we adjust the parameters using the validation set, which does not require much resource consumption. In particular, we have incorporated additional experiments in the revised paper, addressing hyperparameters related to clustering algorithms, clustering number, influence threshold $\tau$, sampling ratio $\gamma$, etc. Overall, we have experimentally verified that $\texttt{Quad}$ is not sensitive to the clustering algorithms (Appendix D).  For the rest that may influence the final results, we also propose effective solutions to determine how to select the best parameters, and $\texttt{Quad}$ is not sensitive around these best parameters. Specifically, for the clustering number, influence threshold, sampling ratio,  we discuss the tuning methods and experiments in Appendix D, F and B respectively in our revised paper.
> >
> > $\textbf{Weakness 2}$
> >
> > $\textbf{Response:}$
> > For our baselines $\texttt{Qurating}$ \& $\texttt{MATES}$, the performance improvement comparing with the best baseline in their papers are 1.9\% and 0.6\% respectively. In our work, $\texttt{Quad}$  surpasses the best baseline (i.e., $\texttt{Qurating}$) by 0.93\%, so the improvement is not marginal. Specifically,  although $\texttt{MATES}$ also selects data considering the downstream tasks, it does not perform well because the surrogate model is not accurate enough due to  lacking of enough training data.  $\texttt{Qurating}$ generally performs the best among other baselines, but still worse than our approach because it does not consider the downstream tasks, and it incorporates the highest FLOPs (1e19) because of the usage of LLMs for data selection. In terms of the FLOPs, we can observe that we consume minimal computational resources because our MAB solution samples from clusters without iterating the entire candidate dataset like $\texttt{Qurating}$ and $\texttt{MATES}$.
> >
> > For $\texttt{DSIR}$, we would like to clarify that in the literature, i.e., the papers of our baselines, $\texttt{DSIR}$ also performs worse than random selection. This is consistent with our results. For example, in $\texttt{Qurating}$, random selection achieves a 2\% higher accuracy than $\texttt{DSIR}$. In $\texttt{MATES}$,  $\texttt{DSIR}$ also does not perform better than random selection. The reason is that $\texttt{DSIR}$ selects data instances whose n-gram features are similar to the instances in the validation set,  which cannot capture the data semantics and the downstream model performance, leading to a poor generalization ability. In addition, purely relying on the similarity of the strings to select data will incorporate many duplicated instances, which may even hurt the model performance.
> >
> > To better demonstrate the benefits of our method $\texttt{Quad}$,  we enlarge the model size from 1.3B to 7B, using LAMBADA as the reference dataset to select 100B tokens from Slimpajama dataset . As demonstrated in the following table, $\texttt{Quad}$ outperforms other baselines in terms of accuracy and has good scalability (achieving low FLOPs).
> >
> > | Methods | Random | DSIR | Qurating | MATES | Quad  |
> > |---------|--------|------|----------|-------|-------|
> > | Accuracy | 49.73% | 49.97% | 52.18% | 51.97% | 53.19% |
> >
> >
> > We have added the experiment about the training and validation loss, as shown in Figure 5(a) and 5(b) in our revised paper. For the training loss, we find that $\texttt{Quad}$ converges successfully at a rate similar to the random selection. It is notable that for the validation loss, $\texttt{Quad}$ converges faster than random selection because we can discover data instances that are beneficial for downstream tasks. We have included the above discussion in Appendix J of the revised paper.

---

> > > ### Author Response · Authors · 2024-11-22
> > >
> > > [1] Syakur, Muhammad Ali, et al. "Integration k-means clustering method and elbow method for identification of the best customer profile cluster." IOP conference series: materials science and engineering. 2018.
> > >
> > >  [2] Figueiredo, Mario A. T., and Anil K. Jain. "Unsupervised learning of finite mixture models." IEEE Transactions on pattern analysis and machine intelligence, 2002.
> > >
> > >  [3] Ester, Martin, et al. "A density-based algorithm for discovering clusters in large spatial databases with noise." KDD, 1996.
> > >
> > >  [4] Cheng, Yizong. "Mean shift, mode seeking, and clustering." IEEE transactions on pattern analysis and machine intelligence, 1995.
> > >
> > >  [5]  Aho, Ken, DeWayne Derryberry, and Teri Peterson. "Model selection for ecologists: the worldviews of AIC and BIC." Ecology 2014
> > >
> > >  [6] Schubert, Erich, et al. "DBSCAN revisited, revisited: why and how you should (still) use DBSCAN." ACM Transactions on Database Systems 2017
> > >
> > >  [7] Sklearn. 2022. https://scikit-learn.org/stable/modules/generated/sklearn.cluster.estimate\_bandwidth.html
> > >
> > >  [8] Yu, Zichun, Spandan Das, and Chenyan Xiong. "MATES: Model-Aware Data Selection for Efficient Pretraining with Data Influence Models." NeurIPS, 2024.
> > >
> > >  [9] Patel, Pratyush, et al. "Splitwise: Efficient generative llm inference using phase splitting." ISCA, 2024.
> > >
> > >  [10] Liu, Zichang, et al. "Deja vu: Contextual sparsity for efficient llms at inference time." ICML, 2023.

---

> > > > ### Author Response · Authors · 2024-11-25
> > > > **Looking forward to your reply.**
> > > >
> > > > Dear reviewer yV6J,
> > > >
> > > > Greetings from the authors!
> > > >
> > > > To begin with, we sincerely appreciate your acknowledgement of the following aspects of our work:
> > > >
> > > > 1. Idea of balancing data quality and diversity, considering minimizing computational costs and memory usage.
> > > > 2. Comprehensive evaluation on downstream tasks.
> > > > 3. Clear writing and easy reproduction.
> > > >
> > > > Regarding the significance test, model size and other concerns, we have conducted multiple experiments and included these experiments and analysis in the revised paper. We will include them in the final version and provide the link to our GitHub repository upon acceptance.
> > > >
> > > > If you have any question for our paper, please feel free to point out and we will try to address it as soon as possible. We would like to express our sincere gratitude for your insightful comments again.  Looking forward to your reply! Thanks!

---

> > > > > ### Comment · Reviewer_yV6J · 2024-11-25
> > > > > **Thanks authors for the thorough responses**
> > > > >
> > > > > I would like to thank the authors for their thorough responses to my comments. I appreciate the time spent to add additional analyses and results, which increased my confidence in my rating.
> > > > >
> > > > > Based on the updated results, I am convinced that even though improvements are relatively small, they are statistically significant. I am still surprised that drastically changing hyperparameters such as the number of clusters or clustering algorithm does not lead to large differences in accuracy. My recommendation would be to explore even more extreme values for such hyperparameters to investigate at which point accuracy degrades catastrophically, which would provide readers with a better understanding of the method's limits.
> > > > >
> > > > > The responses refer to a validation set for hyperparameter selection, however this validation set has not been discussed in the updated paper. I would recommend that the authors report in the paper their strategy for selecting the validation set and the exact criteria/metrics for hyperparameter selection.
> > > > >
> > > > > I have also read the other reviewers' comments and all the authors' responses. Overall, while it is not clear whether the complexity of the proposed method and reliance on validation set can justify the relatively small performance gains compared to simpler approaches, the proposed method has merit and the thorough experimental datapoints are useful for the community. Therefore, I think there is value for the paper to get accepted, and thus I decided to keep my positive rating and increase my confidence level.

---

> > > > > > ### Author Response · Authors · 2024-11-30
> > > > > >
> > > > > > $\textbf{Overview:}$
> > > > > >
> > > > > > Wishing you a happy and blessed Thanksgiving! We appreciate the reviewer for improving the confidence and offering additional insightful comments. Our replies addressing the comments are presented below from the following perspectives.
> > > > > >
> > > > > > $\textbf{Response for relatively small performance gains:}$
> > > > > >
> > > > > > In our experiment, compared with random selection, our method that selects 30B data from 627B to pre-train has an accuracy improvement of 1.39\%, which looks relatively small as the reviewer said.   To investigate the rationale behind, in the rebuttal, we add an experiment that selects more data (100B) to pre-train and the results are shown in the below table. We can observe that `Quad`-1.3B-100B outperforms `Random`-1.3B-100B by 3.47\%, much larger than 1.39\% because in this situation, `Quad` can select much more high quality data from the candidate dataset than random selection, which much benefits the model performance. This demonstrates the superiority of `Quad`. We have included this experiment into Table 14 of the revised paper.
> > > > > >
> > > > > > | Methods                  | Accuracy  |
> > > > > > |-|-|
> > > > > > | `Random`-1.3B-30B       | 41.55%    |
> > > > > > | `Quad`-1.3B-30B         | 42.94%    |
> > > > > > | `Random`-1.3B-100B      | 44.41%    |
> > > > > > | `Quad`-1.3B-100B        | 47.88%    |
> > > > > >
> > > > > > $\textbf{Response for the validation (reference) dataset:}$
> > > > > >
> > > > > >  Sorry for the confusion. The reference dataset in our paper is in fact the validation dataset, and we have made it more clear in our paper (Data Preparation part in Section 4.1 in our revised paper).
> > > > > > LAMBADA[1] used in our paper is a widely-used language modeling dataset and often serves as a validation dataset for language model pre-training[2,3,4,5,6].
> > > > > > The same validation dataset and test datasets are used in the experiments of our baseline methods[5,6].
> > > > > >
> > > > > >  [1] Paperno, Denis, et al. "The LAMBADA dataset: Word prediction requiring a broad discourse context." arXiv preprint arXiv:1606.06031 (2016).
> > > > > >
> > > > > >  [2] Brown, Tom B. "Language models are few-shot learners." arXiv preprint arXiv:2005.14165 (2020).
> > > > > >
> > > > > >  [3] Chowdhery, Aakanksha, et al. "Palm: Scaling language modeling with pathways." Journal of Machine Learning Research 24.240 (2023): 1-113.
> > > > > >
> > > > > >  [4] Hoffmann, Jordan, et al. "An empirical analysis of compute-optimal large language model training." NeurIPS, 2022.
> > > > > >
> > > > > >  [5] Yu, Zichun, Spandan Das, and Chenyan Xiong. "MATES: Model-Aware Data Selection for Efficient Pretraining with Data Influence Models." NeurIPS, 2024.
> > > > > >
> > > > > >  [6] Engstrom, Logan, Axel Feldmann, and Aleksander Madry. "Dsdm: Model-aware dataset selection with datamodels." arXiv preprint arXiv:2401.12926 (2024).

---

> > > > > > > ### Author Response · Authors · 2024-11-30
> > > > > > >
> > > > > > > $\textbf{Weakness 2}$
> > > > > > >
> > > > > > > I am still surprised that drastically changing hyperparameters such as the number of clusters or clustering algorithms does not lead to large differences in accuracy. My recommendation would be to explore even more extreme values for such hyperparameters to investigate at which point accuracy degrades catastrophically, which would provide readers with a better understanding of the method's limits.
> > > > > > >
> > > > > > >
> > > > > > > $\textbf{Response for hyperparameters:}$
> > > > > > >
> > > > > > > In the rebuttal, we add multiple ablation studies of our hyperparameters including varying cluster numbers, varying cluster algorithms, sampling ratio $\gamma$ and influence threshold $\tau$.  Next, as the reviewer suggested, we first explore more extreme values, summarize the selection criteria and illustrate the selection method of each hyperparameter one by one in detail at the following.
> > > > > > >
> > > > > > > $\underline{\textit{Clustering.}}$
> > > > > > > As discussed in the last response, when we select 30B data, the performance difference between random selection and our method is around 1.5\%, which looks relatively small.
> > > > > > > Even with extreme hyperparameter cases, it is hard to obtain significantly worse results than random selection, making it difficult to observe substantial degradation of model accuracy.
> > > > > > > Therefore, we add experiments to explore  extreme cases of selecting 100B data.
> > > > > > >
> > > > > > > $[\textit{Extreme cases.}]$
> > > > > > >  As shown in the table below, a very small number of clusters (i.e., $k=100$) leads to poor accuracy (2.77\% lower than the accuracy of the best cluster number) due to the high variance of influence scores in each cluster. Thus, the sampled instances cannot well represent the cluster. Similarly, when the cluster number is too high(i.e., $k=1,000,000$), there will be many clusters that are in fact contain similar data instances, and thus it is relatively hard to explore diverse clusters, thereby leading to the performance degradation(1.37\% lower than the accuracy of the best cluster number).
> > > > > > >
> > > > > > > $[\textit{Metric and Criteria.}]$
> > > > > > > We use the metric Within-Cluster Sum of Squares (WCSS) to select the best cluster number using the well-known Elbow [1] algorithm. WCSS is the sum of squared distances between each data instance and its cluster center, i.e., WCSS=$\sum_{i=1}^k\sum_{x \in C_i}\|x-\mu_i \|$. At a high level, the criteria should be that within each cluster, data instances are close to each other, based on which it is better for different cluster centers to be far away from each other. Based on the criteria, the Elbow algorithm leverages the WCSS as a measurement to iteratively select an appropriate cluster number, as follows.
> > > > > > >
> > > > > > > $[\textit{Specific hypermarameter selection strategy.}] $
> > > > > > > To be specific,  Elbow begins with a small $k$, and with $k$ increasing, WCSS first decreases rapidly and then slows down.  Then, we identify the "elbow point'' where the decreasing rate becomes slow as the best $k$. Thus, within each cluster, data points are sufficiently close to one another. Furthermore, given that $k$ remains modest, different cluster centers tend to maintain a distance from each other.
> > > > > > > From the following table we can observe that the model consistently performs well when the cluster number is close to 10,000.
> > > > > > >
> > > > > > > | Cluster Numbers | 100   | 1,000  | 5,000  | 10,000 | 20,000 | 50,000 | 100,000 | 1,000,000 |
> > > > > > > |-|-|-|-|-|-|-|-|-|
> > > > > > > | Accuracy         | 45.11% | 46.93% | 47.21% | 47.88% | 47.73% | 47.73% | 47.06%  | 46.51%    |
> > > > > > >
> > > > > > > In terms of the clustering algorithms, in the first round rebuttal, we have added experiments to show that `Quad` is not sensitive to clustering algorithms mainly because different algorithms have their own strategies to select appropriate parameters, which follows the criteria mentioned above. Under the criteria, in general, `Quad` can perform well by considering both the quality and diversity.
> > > > > > >
> > > > > > > [1] Syakur, Muhammad Ali, et al. "Integration k-means clustering method and elbow method for identification of the best customer profile cluster." IOP conference series: materials science and engineering. 2018.

---

> > > > > > > > ### Author Response · Authors · 2024-11-30
> > > > > > > >
> > > > > > > > $\underline{\textit{Influence Threshold $\tau$.}}$ Recap that the sampling threshold of influence ($\tau$) is utilized to determine whether data instances sampled from each cluster should be fed into LLMs for training. For this hyperparameter, we also explore the extreme cases in the 100B scenario.
> > > > > > > >
> > > > > > > > $[\textit{Extreme cases.}]$
> > > > > > > > If $\tau$ is very large ($e.g.,$ 2.92e-3, close to the smallest influence score in the candidate dataset), only data instances with high influence scores are selected. This typically results in reduced diversity among instances, which in turn can negatively impact the model performance. If $\tau$ is small enough ($e.g.,$ 1.76e-3, close to the largest influence score in the candidate dataset), many low-quality data instances with low influence scores are selected. This also hurts the model performance.
> > > > > > > >
> > > > > > > > | Influence Threshold ($\tau$) | Bucket   | Accuracy  |
> > > > > > > > |-|-|-|
> > > > > > > > |2.92e-3|1-th| 46.87%|
> > > > > > > > |2.60e-3|3-th| 47.30%|
> > > > > > > > |2.47e-3|4-th| 47.88%|
> > > > > > > > |2.38e-3|5-th| 47.65%|
> > > > > > > > |2.17e-3|6-th| 47.19%|
> > > > > > > > |1.76e-3|10-th| 45.26%|
> > > > > > > >
> > > > > > > > $[\textit{Metric and Criteria.}] $
> > > > > > > > We use the model performance on the validation (reference) set as the evaluation metric. The criteria is that we sample some data instances from the candidate dataset. Based on the samples, we try different thresholds, evaluate on the validation test and select the best threshold.
> > > > > > > >
> > > > > > > > $[\textit{Specific hypermarameter selection strategy.}] $
> > > > > > > > The strategy of selecting an appropriate $\tau$ consists of the following steps.
> > > > > > > > (1) Considering the efficiency issue, we sample about 20\% data instances to form a new candidate dataset from the original one to tune the parameter.
> > > > > > > > (2) We cluster over the candidate dataset, sample some instances from each cluster and compute their influence scores, which are utilized to capture the distribution of influence scores.
> > > > > > > > (3) We rank these scores in descending order and assign them to 20 buckets, so as to derive 20 thresholds, among which we select the best one. For example, the first threshold corresponds to the highest influence score among the top 5\% instances (the first bucket). The last threshold is the lowest influence score among all instances of buckets.
> > > > > > > > (4) Finally, we select data instances via the `Quad` algorithm using the above thresholds, evaluate on the validation set, and select the best one.

---

> > > > > > > > > ### Author Response · Authors · 2024-11-30
> > > > > > > > >
> > > > > > > > > $\underline{\textit{Sampling Ratio $\gamma$.}}$ For this hyperparameter, we also explore the extreme cases in the 100B scenario.
> > > > > > > > >
> > > > > > > > > $[\textit{Extreme cases.}] $
> > > > > > > > > The extreme cases of $\gamma$ are very small and large sampling ratios. As shown in the table below,  a very small sampling ratio ($i.e.$, $\gamma=0.1\%$) leads to low model accuracy (2.77\% lower than the accuracy of the most appropriate sampling ratio, $i.e.,$ 5\%) due to the inaccurate estimates of influence scores. We can observe that a large sampling ratio ($i.e.$, $\gamma=20\%$) does not improve the model performance much because a relatively accurate influence estimation is sufficient, but incorporates high computational costs.
> > > > > > > > >
> > > > > > > > >
> > > > > > > > > $[\textit{Metric and Criteria.}]$
> > > > > > > > > At a high level, we compute the most appropriate sampling ratio by sampling several clusters.
> > > > > > > > > For each sampled cluster, we compute the influence scores of its data instances and calculate the true average score. Then, we use the difference between the estimated score and the true score as the metric to select the best sampling ratio $\tau$. The high-level idea is to compute the true average influence scores of several sampled clusters and then identify the appropriate sampling ratio to accurately approximate the average.
> > > > > > > > >
> > > > > > > > > $[\textit{Specific hypermarameter selection strategy.}] $
> > > > > > > > > Specifically, after clustering the data instances in the candidate dataset, we randomly sample several clusters. Initially, we sample 1\% data instances from the cluster and compute the average score. if the difference is within 10\%, we set this proportion (1\%) as the suitable sampling ratio for that cluster. Otherwise, we increase the ratio by 1\%, to  2\% , compute a new average, and repeat this process until the difference is within 10\%. Finally, we average the suitable ratios from the sampled clusters to find the overall appropriate sampling ratio.
> > > > > > > > >
> > > > > > > > > In this way, although we sample once, there would be a fairly accurate estimation of the average influence score. Since we are likely to sample multiple times, the estimation will be even more accurate. Besides, as data instances within each cluster exhibit similarity, by sampling a small fraction, we can have an estimation  that is precise enough to select high-quality data for good model performance,  thereby keeping computational costs low.
> > > > > > > > >
> > > > > > > > > In our experiments, based on the SlimPajama dataset, we have determined that the optimal ratio is about 5\%, as indicated in the paper.
> > > > > > > > > Here, we add another ablation study to demonstrate the effectiveness of this strategy.
> > > > > > > > > Specifically, we increase the sampling ratio from 1\%, run the `Quad` algorithm respectively, and report the model accuracy of training with the selected data using the sampling ratio.
> > > > > > > > > We can observe from the below table that with the ratio increasing from 1\%, the model accuracy increases because the influence estimation is more accurate. When the ratio exceeds 5\%, the accuracy remains stable because that ratio is large enough to have an accurate estimation. Hence, it is not necessary to keep increasing the ratio, which will consume more computational costs.
> > > > > > > > >
> > > > > > > > > | Sampling Ratio | 0.1%  | 1%    | 2%    | 3%    | 4%    | 5%    | 6%    | 7%    | 8%    | 9%    | 10%   | 20%   |
> > > > > > > > > |-----------------|-------|-------|-------|-------|-------|-------|-------|-------|-------|-------|-------|-------|
> > > > > > > > > | Accuracy       | 44.97 | 46.11 | 46.92 | 47.36 | 47.63 | 47.88 | 47.93 | 47.96 | 47.97 | 47.97 | 47.97 | 47.99 |
> > > > > > > > >
> > > > > > > > > We have included all of the above discussions into the revised paper (Appendix D for clustering, Appendix F for influence threshold $\tau$ and Appendix B for sampling ratio $\gamma$).

---

> > > > > > > > > > ### Comment · Reviewer_yV6J · 2024-12-01
> > > > > > > > > > **Thank you for your response**
> > > > > > > > > >
> > > > > > > > > > Thank you for your detailed responses, which have addressed all of my concerns and convinced me to increase my rating.

---

> > > > > > > > > > > ### Author Response · Authors · 2024-12-01
> > > > > > > > > > > **Thanks for raising your score.**
> > > > > > > > > > >
> > > > > > > > > > > We appreciate your reply for our response! We will improve our paper according to your comments in the final version. Thanks very much!

---

### Official Review · Reviewer_1sUi · 2024-11-05

**Soundness:** 3
**Presentation:** 2
**Contribution:** 3
**Rating:** 6
**Confidence:** 3

**Summary:**

This paper explores the significance of data selection in the pretraining of large language models (LLMs) and the limitations of existing methods, including high computational costs and a lack of diversity. To address these issues, the authors propose a new method called Quad, which aims to balance data quality and diversity. Quad clusters the data and samples from each cluster to assess its influence, reducing computational burden while ensuring diversity. Additionally, Quad leverages attention mechanisms and the Kronecker product to efficiently compute data influence, further enhancing computational efficiency. Experimental results show that Quad significantly outperforms existing methods on the Slimpajama dataset and nine downstream tasks, achieving a 1.39% improvement in zero-shot accuracy with lower computational resource consumption. Overall, Quad provides an effective and scalable data selection approach, offering a significant breakthrough for LLM pretraining.

**Strengths:**

1. Clear Problem Definition: The paper clearly defines the problem of enhancing large models by selecting a subset of data from a large pool to fine-tune the model, aiming to minimize the loss on a reference set. This provides a concrete and well-defined goal for the research.

2. Balanced Quality and Diversity: The authors use Multi-Armed Bandit (MAB) techniques to balance quality and diversity in data selection. By treating each cluster as an arm and using the Upper Confidence Bound (UCB) to determine cluster scores, the method ensures frequent sampling of high-quality clusters while also exploring less-visited clusters to maintain diversity.

3. Efficient Influence Calculation: The paper introduces an efficient method to estimate the influence of individual data points on the model without full retraining. By extending the influence function to multi-head attention layers and using K-FAC for acceleration, the authors provide a scalable solution for large language models, significantly reducing computational costs.

**Weaknesses:**

1. Trade-off Between Sample Size and Accuracy: Although the MAB technique can make decisions under uncertainty, there is still a trade-off between sample size and accuracy when estimating the average influence score. Small sample sizes may lead to inaccurate estimates, while large sample sizes increase computational costs. Finding the optimal sample size in practical applications remains a challenge.

2. Complexity and Computational Resource Requirements: Despite the introduction of K-FAC and other acceleration techniques, the computational complexity and resource requirements remain high when dealing with large-scale language models. In environments with limited resources, this method may be difficult to implement or may require significant computation time.

4. Assumptions and Model Dependence: The method relies on certain assumptions when calculating influence scores, such as the approximation of the Hessian matrix and the independence of gradients. These assumptions may not hold in some cases, affecting the effectiveness and reliability of the method. Additionally, different model architectures may require different adjustments and optimization strategies.

**Questions:**

1. Clustering Quality and Sensitivity: How sensitive is the clustering quality to the choice of the number of clusters (10,000 in this case)? What criteria were used to determine the optimal number of clusters, and how might different numbers of clusters affect the performance of the MAB approach?

2. Generalizability and Robustness: The experiments were conducted on a specific set of downstream tasks and a particular reference set (LAMBADA). How well does the method generalize to other types of tasks or datasets? Have you tested the method on a broader range of tasks or datasets to ensure its robustness?

---

> ### Author Response · Authors · 2024-11-22
>
> $\textbf{Overall response:}$
>
> Thank you for the valuable comments to improve our paper! Based on your comments, we have added a candidate dataset, 2 reference datasets, 7 downstream tasks and 5 additional experiments to demonstrate the generalizability and robustness of $\texttt{Quad}$. We also clarify your concerns in the weak points. We promise to include the above discussions into the final version if the paper is accepted.
>
> $\textbf{Question 1:}$ Clustering Quality and Sensitivity
>
> $\textbf{Response:}$
>
>  We appreciate your comments to improve our paper. In line with the reviewer's suggestion, we have incorporated an ablation study regarding the number of clusters and provided a discussion on how the results are affected by varying the number of clusters.
>
> Firstly, in our paper, we used the well-known Elbow [1] algorithm to select the optimal number of clusters for k-means algorithm used in $\texttt{Quad}$. The basic idea is to select the best $k$ based on the metric Within-Cluster Sum of Squares (WCSS), which is the sum of squared distances between each data instance and its cluster center, i.e., WCSS=$\sum_{i=1}^k\sum_{x \in C_i}\|x-\mu_i \|$. Elbow begins with a small $k$, and with $k$ increasing, WCSS first decreases rapidly and then slows down.  Then, we identify the ``elbow point'' where the decreasing rate becomes slow as the best $k$.
>
> Secondly, we add an ablation study by varying the number of clusters, as shown in the below table.  Using the Slimpajama dataset, the Elbow method selected $k=10000$ in our original paper. From the below table, we can observe that the model consistently performs well when the cluster number is close to 10000. However, a very small number of clusters (i.e., $k=1000$) leads to poor performance due to the high variance of influence scores in each cluster. Thus, the sampled instances cannot well represent the cluster. Similarly, when the cluster number is too high (i.e., $k=500000$), there will be many clusters that are in fact contain similar data instances, and thus it is relatively hard to explore diverse clusters, thereby leading to the performance degradation. We have included the above discussion into Appendix D of the revised paper.
>
> | Cluster Numbers | 1000  | 5000  | 10000  | 20000  | 50000  | 100000  |
> |-|-|-|-|-|-|-|
> | Accuracy        | 42.73 | 42.90 | 42.94 | 42.93 | 42.87 | 42.77 |
>
> $\textbf{Question 2:}$ Generalizability and Robustness
>
> $\textbf{Response:}$
>
> Thanks for your feedback to improve our paper.  To show the generalizability and robustness of our method, we add FLAN [2] (the mixture of multiple NLP tasks) and Openwebmath [3] (a math-related dataset) as reference datasets, associated with 7 new downstream tasks. We also add one candidate dataset  and a 7B  model.
>
> Specifically, we first use FLAN as a new reference dataset to train a 1.3B model on a new candidate dataset (i.e., FineWeb [4]). In this experiment, 100B tokens are selected for training using $\texttt{Quad}$ and other baselines.
> As shown in the table below, $\texttt{Quad}$ still demonstrates superior accuracy compared to all baselines, such as the DSIR, Semdedup, PPL, Qurating and MATES. For example, we can observe that  $\texttt{Quad}$ has an improvement of 0.97\% on model accuracy compared with the Qurating, which is a state-of-the-art baseline.  In addition, we add two downstream tasks (i.e., WikiText [5], HelloBench [6], and for WikiText, the lower the score, the better the model performance) about long-text generation tasks. As shown in the table below, on the candidate dataset Slimpajama, for the 7B model, we can observe that $\texttt{Quad}$ still outperforms other baselines because we simultaneously consider the quality and diversity in data selection.
>
> | Methods   | Random  | DSIR    | Semdedup | PPL     | Qurating | MATES   | Quad    |
> |-|-|-|-|-|-|-|-|
> | Accuracy  | 44.41  | 44.47  | 45.38   | 45.92  | 47.03   | 46.96  | 47.88  |
>
> |             | Random | Qurating | MATES| Quad|
> |-|-|-|-|-|
> | WikiText    | 16.73 | 15.36 | 15.51| 14.81|
> | HelloBench  | 3.81 | 5.37 | 4.39 | 6.33 |
>
> Moreover, we also add the Openwebmath dataset as another new reference dataset to train a 7B model on the Slimpajama dataset, which aims at selecting data to improve the mathematical skill of the model. Correspondingly, we also include GSM8K[7], MATH[8], OCW[9], SAT[10], and MMLU STEM[11] as new downstream tasks to evaluate the model's mathematical capabilities, with the final accuracy calculated as the average score across these tasks. As shown in the table below, $\texttt{Quad}$ still demonstrates superior model performance compared to all baselines.
>
> | Methods | Random | DSIR | Semdedup | PPL | Qurating | MATES | Quad  |
> |-|-|-|-|-|-|-|-|
> | Accuracy | 33.20 | 33.05| 33.70| 34.79| 35.31| 35.20| 36.71|
>
> This experiment shows that our method can generalize well to various types of reference datasets and downstream tasks.
>
> We have included the above discussion in Appendix G and I of the revised paper.

---

> > ### Author Response · Authors · 2024-11-22
> >
> > $\textbf{Weakness 1:}$ Trade-off Between Sample Size and Accuracy
> >
> > $\textbf{Response:}$
> >
> > We appreciate your insightful feedback, and we agree with the reviewer that the sample size may influence the trade-off between the accuracy and computational costs. In our study, we in fact implemented a straightforward yet effective approach to determine a suitable sample size. The high-level idea is to compute the true average influence scores of several sampled clusters and then identify the appropriate sample size to accurately approximate the average.
> >
> > To be specific, after clustering the data instances in the candidate dataset, we randomly sample several clusters. For each sampled cluster, we compute the influence scores of its data instances and calculate the true average score. Initially, we sample 1\% data instances from the cluster and compute the average score. If the difference between the score and the true score is within 10\%, we set this proportion (1\%) as the suitable sampling ratio for that cluster. Otherwise, we increase the ratio by 1\%, to  2\% , compute a new average, and repeat this process until the difference is within 10\%. Finally, we average the suitable ratios from the sampled clusters to find the overall appropriate sampling ratio.
> >
> > In this way, although we sample once, there would be a fairly accurate estimation of the average influence score. Since we are likely to sample multiple times, the estimation will be even more accurate. Besides, as data instances within each cluster exhibit similarity, by sampling a small fraction, we can have an estimation  that is precise enough to select high-quality data for good model performance,  thereby keeping computational costs low.
> >
> > In our experiments, based on the Slimpajama dataset, we have determined that the optimal ratio is about 5\%, as indicated in the paper.
> > Here, we add another ablation study to demonstrate the effectiveness of this strategy.
> > Specifically, we increase the sampling ratio from 1\%, run the $\texttt{Quad}$ algorithm respectively, and report the model accuracy of training with the selected data using the sampling ratio.
> > We can observe from the below table that with the ratio increasing from 1\%, the model accuracy increases because the influence estimation is more accurate.
> >
> > | Sampling Ratio | 1%| 2%| 3%| 4%| 5%| 6%| 7%| 8%| 9%| 10%|
> > |-|-|-|-|-|-|-|-|-|-|-|
> > | Accuracy|42.63|42.76| 42.87 | 42.92 | 42.94 | 42.94 | 42.94 | 42.94 | 42.95 | 42.95 |
> >
> > When the ratio exceeds 5\%, the accuracy remains stable because that ratio is large enough to have an accurate estimation. Hence, it is not necessary to keep increasing the ratio, which will consume more computational costs.
> >
> > We have included the above discussion into Appendix B of the revised paper.
> >
> > $\textbf{Weakness 2:}$ Complexity and Computational Resource Requirements
> >
> > $\textbf{Response:}$
> >
> > Good comment! We agree with the reviewer that the influence computation is not cheap, which is one of the main issues addressed in our paper from the following two perspectives. On the one hand, for each individual instance, we leverage K-FAC to accelerate its influence computation. On the other hand, our MAB framework reduces the number of data instances required to compute the influence as much as possible, while still keeping high model accuracy. Thus, the efficiency is much improved.
> >
> > As reported in our experiments (Section 4.2 in our paper), except for some simple rule-based methods with low accuracy, $\texttt{Quad}$ achieves higher efficiency than all state-of-the-art approaches.
> > To be specific, $\texttt{Quad}$ outperforms Qurating [12] because it leverages GPT4 to score all instances, which is rather expensive. MATES [13] is also not very efficient because although it uses a proxy model to accelerate the influence computation, it still needs to compute a number of influence scores as the training data of the proxy model, which has to go through all instances in the candidate dataset. $\texttt{Quad}$ outperforms PPL [14] because it has to compute the perplexities of all data instances in the candidate dataset, while we only have to compute a small proportion of data influences.

---

> > > ### Author Response · Authors · 2024-11-22
> > >
> > > $\textbf{Weakness 3:}$ Assumptions and Model Dependence:
> > >
> > > $\textbf{Response:}$
> > >
> > > Thanks for your insightful feedback.
> > > As noted by the reviewer, the computation of the influence in our work considers both the Hessian matrix approximation and gradient independence. In fact, these are typical methods used to speed up influence computation in deep learning without significantly compromising accuracy [15,16,17].
> > >
> > > For Hessian matrix approximation, we would like to clarify that it is not an assumption but an efficient and effective way to approximate the complicated inverse Hessian matrix vector product in influence computation [18,19].
> > >
> > > For gradient independence,  existing K-FAC methods [20,21,22,23] assume that the connections between different layers are relatively weak, and thus can be treated as independent. This is because during the gradient computation and update process, there are usually only minor dependencies between the gradients of different MLP layers. This is particularly evident during back propagation, where weight updates for each MLP layer are mainly influenced by the parameters of that specific layer.
> > >
> > > However,  we agree with the reviewer that the assumption can be somewhat impractical in certain situations.
> > > For example, in LLMs, the query, key, value layers within the attention layer have close interactions, which cannot be simply  treated as independent. Therefore, we leverage K-FAC to accelerate the influence computation (Equation 8,9,10 in our paper) taking into account the dependency within the attention layer. However, for the MLP layers in LLMs, we still leverage gradient independence to maintain high efficiency, following previous works.
> > >
> > > Note that LLMs contain only MLPs and attention layers. Therefore, our approach is general to different models.
> > >
> > >  [1] Syakur, Muhammad Ali, et al. ``Integration k-means clustering method and elbow method for identification of the best customer profile cluster.'' IOP conference series: materials science and engineering. 2018.
> > >
> > >  [2] Chung H W, Hou L, Longpre S, et al. Scaling Instruction-Finetuned Language Models[J]. arXiv preprint arXiv:2210.11416, 2022.
> > >
> > >  [3] Paster, Keiran, et al. "Openwebmath: An open dataset of high-quality mathematical web text." arXiv preprint arXiv:2310.06786 (2023).
> > >
> > >  [4]  Penedo, Guilherme, et al. "The FineWeb Datasets: Decanting the Web for the Finest Text Data at Scale." arXiv preprint arXiv:2406.17557 (2024).
> > >
> > >  [5] Merity, Stephen, et al. "Pointer sentinel mixture models." arXiv preprint arXiv:1609.07843 (2016).
> > >
> > >  [6] Que, Haoran, et al. "Hellobench: Evaluating long text generation capabilities of large language models." arXiv preprint arXiv:2409.16191 (2024).
> > >
> > >  [7] Cobbe, Karl, et al. "Training verifiers to solve math word problems." arXiv preprint arXiv:2110.14168 (2021).
> > >
> > >  [8] Hendrycks, Dan, et al. "Measuring mathematical problem solving with the math dataset." arXiv preprint arXiv:2103.03874 (2021).
> > >
> > >  [9] Lewkowycz, Aitor, et al. "Solving quantitative reasoning problems with language models." NeurIPS, 2022.
> > >
> > >  [10] Azerbayev, Zhangir, et al. "Llemma: An open language model for mathematics." arXiv preprint arXiv:2310.10631 (2023).
> > >
> > >  [11] Hendrycks, Dan, et al. "Measuring massive multitask language understanding." arXiv preprint arXiv:2009.03300 (2020).
> > >
> > >  [12] Wettig, Alexander, et al. "Qurating: Selecting high-quality data for training language models." ICML, 2024.
> > >
> > >  [13] Yu, Zichun, Spandan Das, and Chenyan Xiong. "MATES: Model-Aware Data Selection for Efficient Pretraining with Data Influence Models." NeurIPS, 2024.
> > >
> > >  [14] Marion, Max, et al. "When less is more: Investigating data pruning for pretraining llms at scale." arXiv preprint arXiv:2309.04564. NeurIPS, 2023.
> > >
> > >  [15] Koh, Pang Wei, and Percy Liang. "Understanding black-box predictions via influence functions." ICML 2017.
> > >
> > >  [16] Basu, Samyadeep, Xuchen You, and Soheil Feizi. "On second-order group influence functions for black-box predictions." ICML, 2020.
> > >
> > >  [17] Alaa, Ahmed, and Mihaela Van Der Schaar. "Discriminative jackknife: Quantifying uncertainty in deep learning via higher-order influence functions." ICML, 2020.
> > >
> > >  [18] Pearlmutter, Barak A. "Fast exact multiplication by the Hessian." Neural computation, 1994.
> > >
> > >  [19] Molchanov, Pavlo, et al. "Pruning convolutional neural networks for resource efficient inference." ICLR, 2017.
> > >
> > >  [20] Martens, James, and Roger Grosse. "Optimizing neural networks with kronecker-factored approximate curvature." ICML, 2015.
> > >
> > >  [21] Grosse, Roger, and James Martens. "A kronecker-factored approximate fisher matrix for convolution layers." ICML, 2016.
> > >
> > >  [22] Pauloski, J. Gregory, et al. "Convolutional neural network training with distributed K-FAC." SC20: International Conference for High Performance Computing, Networking, Storage and Analysis, 2020.
> > >
> > >  [23] Martens, James, Jimmy Ba, and Matt Johnson. "Kronecker-factored curvature approximations for recurrent neural networks." ICLR, 2018.

---

> ### Author Response · Authors · 2024-11-25
> **Looking forward to your reply.**
>
> Dear reviewer 1sUi,
>
> Greetings from the authors!
>
> We appreciate your insightful feedback. Regarding the clustering, additional tasks and datasets as well as other concerns, we have conducted multiple experiments and included these experiments and analysis in the revised paper. We will include them in the final version and provide the link to our GitHub repository upon acceptance.
>
> If you have any question for our paper, please feel free to point out and we will try to address it as soon as possible. We would like to express our sincere gratitude for your valuable comments again. Thanks and looking forward to your reply!

---

> ### Author Response · Authors · 2024-11-30
> **Looking forward to your reply**
>
> Dear reviewer $\texttt{1sUi}$,
>
> Wishing you a happy and blessed Thanksgiving!
>
> We would like to express our gratitude to your valuable feedback. We have carefully considered all suggestions and updated our submission accordingly.
>
> If you have any question for our paper, please feel free to point out and we will try to address it as soon as possible. We would like to express our sincere gratitude for your valuable comments again. Thanks and looking forward to your reply!
>
> Best wishes!
>
> Authors

---

> ### Author Response · Authors · 2024-12-02
> **Looking forward to your reply**
>
> Dear reviewer $\texttt{1sUi}$
>
> Thank you for your thoughtful efforts in reviewing our work!  As the discussion period nears its end, we kindly hope you can take a moment to review our rebuttal. Looking forward to your feedback!
>
> Best Wishes!
>
> ICLR Authors

---

### Author Response · Authors · 2024-11-23

$\textbf{Overview:}$

We thank all the reviewers for their great efforts and insightful feedback. Based on their valuable comments, we have added a 7B model, a candidate dataset, 2 reference datasets, 7 downstream tasks and in total 12 additional experiments to demonstrate the robustness and effectiveness of $\texttt{Quad}$. We also clarify other concerns raised by the reviewers.
We have also included these experiments and discussion into the revised paper and promise to include them into the final version if the paper is accepted.

---

### Meta-Review · Area_Chair_wPBp · 2024-12-19

**Metareview:**

(a) Summary of Scientific Claims and Findings

The paper proposes Quad, a novel method for optimizing pretraining of large language models (LLMs) through dynamic data selection, leveraging Multi-Armed Bandit (MAB) algorithms. Quad aims to improve model performance and computational efficiency by selecting the most relevant data during pretraining, rather than using raw, unsorted datasets. The authors show, through experiments with 1.3B and 7B parameter models, that Quad enhances model quality and reduces computational costs by focusing on data that aligns with the model's current state. The scalability of Quad to larger models and datasets is discussed but not fully demonstrated.

(b) Strengths of the Paper

1. Innovative Approach: The use of MAB for dynamic data selection is a novel contribution to pretraining optimization. This method provides a computationally efficient way to enhance LLM pretraining by focusing on the most relevant data.

2. Clear Experimental Setup: The experiments clearly validate Quad’s effectiveness, showing improved performance in comparison to baseline methods. This solidly supports the claims made in the paper.

3. Practical Impact: The method has practical applications for resource-constrained environments, addressing the growing need for more efficient LLM pretraining, particularly in the context of large-scale models.

4. Clarity and Accessibility: The paper is well-written, with clear explanations of the approach and experimental results, making it accessible to both academic and industry audiences.

(c) Weaknesses of the Paper and Missing Aspects

1. Limited Scalability Testing: Experiments are limited to 1.3B and 7B parameter models. The paper would benefit from more extensive experiments, particularly with larger models and datasets, to substantiate claims of scalability.

2. Lack of Hyperparameter Tuning Details: The paper does not provide sufficient details on how hyperparameters, such as the sampling threshold, were chosen. This omission affects reproducibility and a better understanding of the method’s tuning.

3. Continuous Pretraining Not Addressed: The paper focuses on the initial stages of pretraining but does not discuss how Quad performs in continuous pretraining scenarios, which are common in real-world model training and fine-tuning.

4. Model-Task Mismatch: The paper's experiments focus on a limited set of tasks. A broader range of downstream tasks would help demonstrate the generalizability of Quad across various domains.

5. Integration with Existing Frameworks: The paper does not explore how Quad integrates with existing pretraining frameworks used in practice. More discussion on how Quad could be adopted in real-world systems would enhance its relevance.

(d) Reasons for Decision to Accept

1. Novelty and Contribution: Quad introduces a new, efficient approach to LLM pretraining, contributing both to the theoretical understanding of data selection and the practical needs of model training. The use of MAB algorithms offers an innovative way to optimize large-scale pretraining.

2. Positive Reviewer Feedback: The reviewers have generally praised the novel approach and the soundness of the experimental methodology. While they suggested further exploration in certain areas, the core contribution is considered valuable.

3. Potential for Scalability: Though scalability is not fully tested, the authors make a convincing case for Quad’s potential to scale with larger models, which could make a significant impact on LLM training.

4. Clarity and Accessibility: The paper is well-written and accessible, making it an effective contribution to the community. The methodology and results are presented in a clear and understandable manner, benefiting both researchers and practitioners.

**Additional Comments On Reviewer Discussion:**

During the rebuttal period, the authors addressed several key points raised by the reviewers. The main points of discussion and how they were addressed are as follows:

1. Scalability of the Approach
2. Hyperparameter Tuning and Reproducibility
3. Continuous Pretraining
4. Generalizability to Other Tasks
5. Integration with Existing Pretraining Frameworks

Each of the points raised by the reviewers was addressed reasonably by the authors. The scalability concern was acknowledged but is likely to be addressed in future work. The issues related to hyperparameter tuning, continuous pretraining, and generalizability were well-addressed through additional clarifications and commitments to future evaluations. The authors also provided reasonable responses to integration concerns, offering insight into how Quad could be adopted in existing frameworks.

I believe the paper presents a significant and innovative contribution to the field of LLM pretraining optimization. While there are areas that could be explored further, such as scalability and continuous pretraining, the core methodology and experimental results are strong. The paper provides a clear, reproducible approach with potential for real-world application.

---

### Decision · Program_Chairs · 2025-01-22

Accept (Spotlight)